# What Formal Language Classes Can Neural Networks Learn? An Empirical Study

## Abstract

Understanding neural networks' algorithmic capabilities is key to predicting their performance on real-world tasks, and formal language recognition offers a rigorous framework for evaluating what algorithms these models can employ. To this end, a growing body of theoretical work has sought to characterize the classes of formal languages that various neural network architectures—particularly transformers—can recognize. These mathematical results generally rely on simplified versions of the transformer network that differ from those used in practice, so empirically testing the hypothesized expressivity bounds remains crucial. Some work has already evaluated neural networks as recognizers of certain representative languages, but in order to test broad learnability claims, one should test on an appropriately broad distribution of languages and see how well the hypotheses and experiments align. This paper is an attempt to do just that, using efficient algorithms for randomly sampling formal languages from certain language classes and generating labeled datasets to be used for training and evaluating neural networks on those languages. We do so for the classes of regular languages, context-free languages, partially-ordered star-free languages, and star-free languages of varying dot-depth. Along the way, we develop a novel algorithm for efficiently sampling a string of a desired length from a probabilistic context-free grammar. We experimentally test the language recognition performance of transformers, simple RNNs, and LSTMs on these classes and find that transformers can recognize only a subset of star-free languages. On context-free languages, we find that all models learn very strong approximations of the languages, but fail to learn the correct underlying mechanism. Our code is publicly available.[1]

## 1 Introduction

As neural networks power an ever-expanding range of applications, understanding their algorithmic capabilities is essential for predicting real-world performance. A central question is whether they can learn the algorithms required for complex tasks—ranging from identifying structural patterns to verifying logical consistency. Formal language recognition provides a precise framework for addressing this question, allowing us to characterize precisely the computational abilities of neural networks. A model that recognizes a particular class of formal languages has demonstrably learned the corresponding class of algorithms. This perspective provides a principled approach to assessing neural networks' computational limits and, as networks are deployed for increasingly complex reasoning tasks, offers crucial insight into the algorithmic processes they can systematically learn and execute across diverse inputs.

Numerous theoretical results have characterized the formal capabilities of models such as RNNs and transformers (see, for example, the survey of Korsky & Berwick (2019) for RNNs and Strobl et al. (2024) for transformers). Yet, these theoretical analyses almost always rely on highly idealized assumptions that do not hold in realistic training scenarios—for instance, replacing softmax with hard attention (Yang et al., 2024; Merrill et al., 2022; Barceló et al., 2024), assuming infinite-precision arithmetic (Barceló et al., 2024; Merrill & Sabharwal, 2023) or non-standard positional encodings that lack empirical validation (Pérez et al., 2019; Chiang & Cholak, 2022). As a result, they provide at best an upper bound on what might be possible, rather than a reliable account of what networks

---

[1]See the anonymous supplementary material.

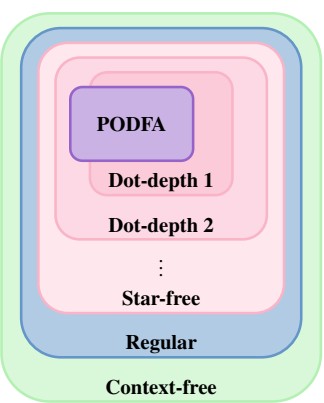

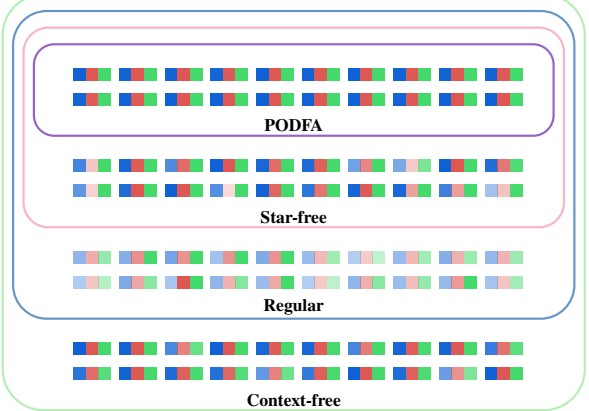

(a) The star-free languages are the union of star-free languages of any finite dot-depth. PODFA languages are a strict subset of the star-free languages of dot-depth 2, but are incomparable to star-free languages of dot-depth 1. Regular languages are a strict superset of the star-free languages, and context-free languages are a strict superset of the regular languages.

(b) Summary of the results on each of the 20 languages from each language class. For each language, we show the accuracy of the best-performing model for transformers (blue), RNNs (red), and LSTMs (green). The opacity of each rectangle is chosen based on the model performance, i.e., an opaque rectangle indicates high accuracy, while a more transparent rectangle indicates a low accuracy. Note that because our language sampling algorithms do not guarantee non-membership in a lower class, each language may in fact belong to a smaller bubble than the one in which it is shown.

Figure 1: Summary of language classes that we use in this paper, together with a summary of the performance achieved by transformers, RNNs, and LSTMs on each language from these language classes. All architectures achieve near-perfect accuracy on PODFA languages, but rarely achieve perfect accuracy on more general star-free languages. All architectures perform poorly on regular languages, but they perform surprisingly well on context-free languages.

can actually learn in practice. To bridge this gap, empirical studies have played a central role, typically training neural networks on carefully chosen languages representative of particular classes (Delétang et al., 2023; Butoi et al., 2025). However, focusing on a few representative hand-picked examples, such as PARITY or DYCK-K, risks introducing bias and may yield an incomplete or even misleading picture. To date, no work has systematically assessed neural networks on randomly sampled languages across multiple well-defined classes.

We address this gap by evaluating RNNs, LSTMs, and transformers on a broad set of randomly sampled languages from multiple classes of the Chomsky hierarchy, focusing on regular, star-free, and context-free languages—all commonly used in theoretical analyses of neural network expressivity. Figure 1a shows a summary of the language classes that we focus on in this paper. Transformers with hard attention, no positional embeddings, and strict future masking express exactly the star-free languages (Yang et al., 2024). In contrast, transformers with soft attention, strict future masking, and no positional embeddings can express only partially-ordered star-free languages (Li & Cotterell, 2025), which are a proper subset of the star-free languages. Unlike prior work that focuses on a few representative languages, our approach offers a more robust assessment of neural networks' recognition capabilities by sampling systematically from these language classes. This broader sampling is theoretically motivated: the contrast between partially-ordered and general star-free languages provides a natural testbed for analyzing the expressivity gap between soft- and hard-attention transformers, and sampling across both families allows us to analyze empirically whether these theoretical limits manifest in practice. Similarly, while DYCK-K languages are widely used to illustrate hierarchical structure, they are only a narrow subclass of context-free languages, whereas, for instance, real programming languages rely on the full expressive power of general context-free languages.

To facilitate this evaluation, we present sampling algorithms for regular (§4.1.1), partially-ordered star-free and star-free of varying dot-depth (§4.1.2), and context-free languages (§4.1.3). Additionally,

we extend the length-constrained sampling algorithm of Butoi et al. (2025) for context-free languages (§4.2.1) to generate efficiently samples of some desired length.

Our results highlight both the strengths and limitations of neural architectures across several language classes. All models achieve near-perfect recognition accuracy on a subset of star-free languages, which we term partially-ordered, but performance drops sharply on star-free languages of varying dot-depth. For transformers, this contrasts with some theoretical results claiming that they can express all star-free languages. Consistent with theory, transformers fail to learn some regular languages, though LSTMs sometimes succeed. On context-free languages, all architectures achieve high accuracy by learning very good approximations of the target language. However, we show that they fail to learn the correct string generation mechanism. Figure 1b summarizes our results.

## 2 RELATED WORK

Much research has investigated the ability of neural networks to represent and learn formal languages. Broadly, this work falls into two categories: theoretical studies, which examine the representational capacity (Strobl et al., 2024; Yang et al., 2024; Jerad et al., 2025) and learnability (Hahn & Rofin, 2024) of neural networks, and empirical studies (Valvoda et al., 2022; Delétang et al., 2023; Butoi et al., 2025), which test models on specific formal languages or language classes. In this work, we focus on the latter.

Early empirical studies primarily examined whether neural networks could learn particular languages or syntactic patterns. With the advent of large language models in natural language processing, interest in the formal-language learning abilities of neural architectures has resurged, leading to more systematic evaluations across diverse language classes.

Several studies and benchmarks have focused on regular and subregular languages, but these generally did not evaluate multiple neural network architectures on a wide range of randomly generated languages, spanning multiple subclasses that are relevant for expressivity analysis. Valvoda et al. (2022) tested attention-augmented LSTMs on randomly generated finite-state transduction tasks, finding that performance degrades as the underlying automaton grows in size. Similarly, Borenstein et al. (2024) studied RNNs and transformers on probabilistic DFAs, showing decreasing performance with larger state sets and alphabets. In contrast, we test transformers, RNNs, and LSTMs as recognizers, a task that aligns naturally with the Chomsky hierarchy (Butoi et al., 2025), on both randomly sampled regular languages, as well as multiple subclasses of the star-free languages. Van der Poel et al. (2024) introduced a benchmark for regular and subregular languages and reported that GRUs consistently outperform other architectures, including transformers. Despite testing on many subclasses of the regular languages, their work focused on neither the dot-depth hierarchy nor partially-ordered star-free languages. Bhattamishra et al. (2020) demonstrated that transformers fail to recognize even certain star-free languages, specifically those beyond dot-depth 1, by testing on a few representative examples. Similarly, Li & Cotterell (2025) showed that transformers with soft attention cannot learn star-free languages that fall outside of a small fragment of linear temporal logic, which is equivalent to what we call partially-ordered star-free languages languages. Both of these papers evaluate transformers on only a handful of handpicked examples. In contrast, our work provides a more systematic analysis by testing many partially-ordered and general star-free languages across various dot-depths.

On context-free and mildly context-sensitive languages, similar limitations arise. Someya et al. (2024) designed linguistically-motivated test languages spanning different levels of the Chomsky hierarchy and observed that performance decreases with the complexity of the language, and all tested architectures—LSTMs, stack-augmented RNNs, and transformers—fail to learn context-sensitive languages. In a broader study, Delétang et al. (2023) compared standard and memory-augmented models across 15 transduction tasks, showing that transformers underperform even on some regular languages, while memory-augmented models succeed on certain context-sensitive ones. Extending this line of work, Butoi et al. (2025) recasted these tasks as language recognition and confirmed that transformers fail even on many regular languages, whereas RNNs and LSTMs can reliably handle most regular languages, but fail to generalize to higher levels of the hierarchy. In contrast to our work, all of these papers focused on only a few examples of languages from each language class. Khalighinejad et al. (2023) showed that transformers approximate the CKY algorithm on probabilistic context-free grammars, but their accuracy decreases with grammar ambiguity. While

they do sample random context-free languages, their sampling method does not ensure full support over the context-free languages.

## 3 BACKGROUND

An **alphabet** $\Sigma$ is a finite, non-empty set of **symbols**. A **string** over some alphabet $\Sigma$ is a sequence of symbols from $\Sigma$. The Kleene closure $\Sigma^*$ of an alphabet $\Sigma$ is the set of all strings over $\Sigma$. We denote by $\varepsilon$ the empty string. Finally, a **formal language** $L$ over $\Sigma$ is a possibly infinite subset of $\Sigma^*$, and a **language class** is a possibly infinite set of formal languages. Throughout this paper, we use several computational devices that either *recognize* or *generate* formal languages. Regardless of the type of device, we use $w \in L$ to denote that the string $w$ is in the language $L$. We call the proposition $w \in L$ the **label** of $w$.

### 3.1 REGULAR LANGUAGES

**Definition 1.** *A **deterministic finite automaton (DFA)** is a tuple $\mathcal{A} = (Q, \Sigma, \delta, q_0, F)$ where (1) $Q$ is a finite set of **states**, (2) $\Sigma$ is an alphabet, (3) $\delta \colon Q \times \Sigma \to Q \cup \{\emptyset\}$ is the **transition function**, where $\emptyset$ indicates the absence of a transition, (4) $q_0 \in Q$ is the **start state**, and (5) $F \subseteq Q$ is the set of **accept states**. If $\delta(q, a) = r \neq \emptyset$, we say that $\mathcal{A}$ has a transition from $q$ to $r$ that scans $a$, and we write $q \xrightarrow{a} r \in \delta$.*

A sequence $\pi$ of connecting states and transitions of a DFA is called a **path**. If $\pi$'s transitions scan the symbols $w_1, \ldots, w_n$, then we say that $\pi$ **scans** the string $w = w_1 \cdots w_n$. If $\mathcal{A}$ has a path scanning $w$, we say that $\mathcal{A}$ **accepts** $w$; otherwise, we say that $\mathcal{A}$ **rejects** $w$. Moreover, we say that $\mathcal{A}$ **recognizes** the language $\{w \in \Sigma^* \mid \mathcal{A} \text{ accepts } w\}$. Importantly, a language is **regular** if there is a DFA that recognizes it. If all states in a DFA are reachable from the start state and can lead to an accept state, we call it **trim**. We give a more detailed exposition of DFAs in App. A.2.

Regular languages can further be divided into subclasses of subregular languages based on their structural properties. Here, we focus on **star-free** and **partially-ordered** languages,[2] both of which have been used to characterize the expressivity of transformers. For instance, Yang et al. (2024) showed under certain assumptions transformers can express exactly the class of star-free languages, while Jerad et al. (2025) proved that a restricted variant of transformers is equivalent in power to partially-ordered languages.

**Definition 2.** *Let $\delta^* \colon Q \times \Sigma^* \to Q \cup \{\emptyset\}$ be the transitive closure of $\delta$, defined as*

$$\delta^*(q, a) = \delta(q, a), \text{ for } a \in \Sigma \tag{1a}$$

$$\delta^*(q, w_1 \cdots w_n) = \delta(\delta^*(q, w_1 \cdots w_{n-1}), w_n) \tag{1b}$$

*with $\delta^*(q, \varepsilon) = q$ and $\delta^*(\emptyset, w) = \emptyset$ for any $w \in \Sigma^*$.*

*A **partially-ordered DFA (PODFA)** is a DFA $\mathcal{A} = (Q, \Sigma, \delta, q_0, F)$ where there is a partial order relation $\preceq$ on $Q$ defined as $q \preceq r$ if and only if $\delta^*(q, w) = r$ for some string $w \in \Sigma^*$.*

Intuitively, PODFAs are *acyclic* DFAs with possible self-loops, resulting in a partial order over the states. We call partially-ordered languages those languages recognized by partially-ordered DFAs. Importantly, any partially-ordered language is star-free. The star-free languages are most often described by their closure properties.

**Definition 3.** *Star-free languages are the closure of $\emptyset$, $\{\varepsilon\}$, and $\{a\}$ for $a \in \Sigma$, under the operations of union, concatenation, and complementation.*

Intuitively, star-free languages are those that can be described by *extended* regular expressions[3] *without* using the Kleene star operation. Importantly, the language $(ab)^*$ *is* star-free since it can be expressed by the equivalent regular expression $b\overline{\emptyset} \cup \overline{\emptyset}a \cup \overline{\emptyset}aa\overline{\emptyset} \cup \overline{\emptyset}bb\overline{\emptyset}$.

---

[2]While *star-free* is an established term for the class, partially-ordered is not; we introduce it here since we focus on the automata-theoretic characterization of the class.

[3]Extended regular expressions allow complement.

The dot-depth hierarchy is a classification of the star-free languages based on how concatenation and the Boolean operations are nested. The class of star-free languages is the union of the star-free languages of all dot-depths (Cohen & Brzozowski, 1971).

**Definition 4.** *Let $\Sigma$ be an alphabet. Then the languages $\emptyset$, $\{\varepsilon\}$, $\{a\}$, where $a \in \Sigma$ are called **basic languages**.*

We denote by $\mathcal{E}$ the class of basic languages for some alphabet $\Sigma$. Given a class $\mathcal{C}$ of languages, let $\mathrm{B}(\mathcal{C})$ be its Boolean closure, and $\mathrm{M}(\mathcal{C})$ its closure under concatenation.

**Definition 5.** *Let $\mathcal{B}_0 \stackrel{\text{def}}{=} \mathrm{B}(\mathrm{M}(\mathcal{E}))$. The **dot-depth** hierarchy is the sequence obtained from $\mathcal{B}_0$ by setting $\mathcal{B}_{n+1} \stackrel{\text{def}}{=} \mathrm{B}(\mathrm{M}(\mathcal{B}_n))$ for all $n \geq 0$.*

## 3.2 Context-free languages

**Definition 6.** *A **context-free grammar (CFG)** is a tuple $\mathcal{G} = (V, \Sigma, R, S)$ where (1) $V$ is a finite set called the **variables**, (2) $\Sigma$ is a finite set, disjoint from $V$, called the **terminals**, (3) $R$ is a finite set of **rules** of the form $X \to \alpha$, where $X \in V$ and $\alpha \in (V \cup \Sigma)^*$, and (4) $S \in V$ is the **start variable**.*

We call a string $\alpha \in (V \cup \Sigma)^*$ of terminals and variables a **sentential form**. Starting from the start variable, a CFG generates strings by repeatedly applying rules to its variables until it produces a sequence of symbols. If $r = X \to \alpha \in R$ is a rule, and $\alpha_1, \alpha_2 \in (V \cup \Sigma)^*$ are sentential forms, we say that $\alpha_1 X \alpha_2$ **derives** $\alpha_1 \alpha \alpha_2$ in one step, and we write $\alpha_1 X \alpha_2 \stackrel{r}{\Rightarrow} \alpha_1 \alpha \alpha_2$. We call **partial derivation** a sequence of sentential forms and rules $d = \alpha_0 \stackrel{r_1}{\Rightarrow} \alpha_1 \stackrel{r_2}{\Rightarrow} \alpha_2 \ldots \stackrel{r_m}{\Rightarrow} \alpha_m$, where $\alpha_0 = X \in V$ and $\alpha_m = \boldsymbol{w} \in \Sigma^*$. If $\alpha_0 = S$, we call it a **derivation**. We write $\alpha_0 \stackrel{*}{\Rightarrow} \alpha_m$ to denote that $\alpha_0$ **derives** $\alpha_m$ in multiple steps. We call $X \stackrel{*}{\Rightarrow} \boldsymbol{w}$ a partial derivation of $\boldsymbol{w}$ or an $X$-derivation.

A grammar **generates** a string $\boldsymbol{w}$ if $S \stackrel{*}{\Rightarrow} \boldsymbol{w}$. In general, there might be multiple derivations of a string $\boldsymbol{w}$. A language is **context-free** if there is a context-free grammar that generates it. We say that a grammar $\mathcal{G}$ **generates** the language $\{\boldsymbol{w} \in \Sigma^* \mid \mathcal{G} \text{ generates } \boldsymbol{w}\}$. A grammar is called **trim** if all its variables can both be derived from the start symbol and can derive a (possibly empty) sequence of terminals. See App. A.3 for a formal definition and an algorithm for trimming a CFG.

**Definition 7.** *A CFG $\mathcal{G} = (V, \Sigma, R, S)$ is in **Chomsky normal form (CNF)** if $R$ only contains rules of the form (1) $S \to \varepsilon$; (2) $X \to YZ$, where $Y, Z \in V \setminus \{S\}$; or (3) $X \to a$, where $a \in \Sigma$.*

## 4 Method

We propose a method for training neural networks as recognizers of *random* languages across different levels of the Chomsky hierarchy. Unlike most prior work that focuses on handpicked languages, we address challenges such as generating random languages that tend to not belong to lower classes, and length-constrained sampling. We present algorithms for randomly sampling regular, star-free, and context-free languages. We reuse the length-constrained sampling method of Butoi et al. (2025) for regular and star-free languages, and give a novel extension for context-free languages.

### 4.1 Language sampling

A key contribution of our paper is a set of sampling strategies that ensure: (1) the sampling distribution has full support over the target (sub)class while excluding languages from lower levels of the Chomsky hierarchy, and (2) the sampled languages are unlikely to degenerate into a lower (sub)class. Our methods ensure that evaluation reflects the inherent diversity of the language class, and that avoid degenerate cases, on which evaluation would no longer reflect the intended complexity of the target class. We provide algorithms for sampling regular, star-free, and context-free languages that satisfy these conditions.

### 4.1.1 SAMPLING REGULAR LANGUAGES

Several past papers have proposed methods of randomly sampling regular languages (Valvoda et al., 2022; Borenstein et al., 2024; van der Poel et al., 2024; Akyürek et al., 2024). Unlike these methods, we also ensure that the distribution we sample from has full support over the regular languages, which avoids sampling bias toward specific subclasses or structural patterns. First, we sample the number of states and alphabet size from a negative binomial distribution. Then, for each state and symbol, we add a transition with some predefined probability, where the destination state is sampled uniformly at random. Similarly, we make a state final with some predefined probability. Finally, we trim the generated DFA, as required by the length-constrained sampling algorithm of Butoi et al. (2025). Because the number of states and alphabet size are sampled from a distribution with full support over positive integers, it is possible to include any transition and make any state final, this procedure ensures full support over the set of all possible DFAs, and therefore the regular languages. We give the full algorithm and some additional details in App. B.1.

### 4.1.2 SAMPLING STAR-FREE LANGUAGES

To sample PODFAs, we use a similar algorithm to the one described in §4.1.1. Here, we fix an ordering of the states. For each state and input symbol, when we add an outgoing transition, we choose its destination uniformly at random from the states that come later in the ordering. See App. B.2 for the full algorithm.

We also devise a method for sampling star-free languages whose dot-depth is bounded by some constant $K$. To the best of our knowledge, this is the first study that assesses the performance of neural networks on languages of different dot depths. Notice that in general, computing the dot-depth of a star-free language is NP-complete, therefore our algorithm only guarantees that the dot-depth of the sampled automaton is upper bounded by a constant. The key idea is to generate new automata by alternating between applying concatenation and Boolean operations, starting from simple automata encoding the basic languages. We give the full algorithms and some further explanations in App. B.3.

### 4.1.3 SAMPLING CONTEXT-FREE LANGUAGES

We sample random context-free languages via random CFGs in Chomsky normal form. This enables efficient membership checking using standard algorithms such as CKY (Cocke, 1969; Kasami, 1965; Younger, 1967). We sample CFGs with a certain number of terminals, variables, lexical rules, and binary rules (Clark & Fijalkow, 2021; Khalighinejad et al., 2023). Unlike in previous work, we sample these numbers from a negative binomial distribution, which ensures full support over all context-free languages. We then sample uniformly without replacement the lexical and binary rules from the set of all possible rules.

Naïvely sampling CFGs may yield regular languages, which is undesirable. While checking whether a CFG defines a regular language is undecidable (Greibach, 1968), the following characterization of non-regular context-free derivations (Chomsky, 1959) allows us to reduce this likelihood.

**Definition 8.** *A derivation is called **self-embedding** if it has the form $X \overset{*}{\Rightarrow} \boldsymbol{w}_1 X \boldsymbol{w}_2$, where $\boldsymbol{w}_1, \boldsymbol{w}_2 \in \Sigma^*$. A CFG is called **self-embedding** if it has at least one self-embedding derivation. A language is called **self-embedding** if all CFGs that generate it are self-embedding.*

Importantly, all non-regular context-free languages are self-embedding (Chomsky, 1959), so we introduce self-embedding derivations into our sampled CFGs. This does not guarantee non-regularity, however, since we only sample one CFG encoding the language.[4] We give the full algorithms and additional details about sampling context-free languages in App. B.4.

## 4.2 DATASET GENERATION

We reuse the general method devised by Butoi et al. (2025) to generate a dataset of $N$ examples with lengths in $[n_{\min}, n_{\max}]$ for language $L$. We give a short summary here for completeness. We uniformly sample a positive or negative label $N$ times, then we generate an example accordingly. Positive strings are produced using class-specific algorithms: for regular and subregular languages,

---

[4]Any language can have many CFGs encoding it, some of which might not be self-embedding.

we reuse Butoi et al.'s (2025) algorithm, and for context-free languages we give a novel length-constrained sampling algorithm based on their method (§4.2.1). Negative strings are generated with equal probability as random or adversarial examples. Random negative examples are sampled uniformly from $\Sigma^n$, where the length $n$ is uniformly sampled from $[n_{\min}, n_{\max}]$. We generate adversarial examples by first sampling a positive example of some uniformly sampled length $n$, then applying $K$ random edits, where $K$ follows a geometric distribution favoring small values. Each candidate negative is verified with class-specific algorithms; if found positive, the process restarts.

### 4.2.1 LENGTH-CONSTRAINED SAMPLING FROM CONTEXT-FREE LANGUAGES

To allow string sampling from its language, a CFG must define a probability distribution. A CFG's productions can be weighted using weights from a semiring, resulting in a CFG that generates strings with some weight. If these weights form a probability distribution, we can use them for string sampling.

**Definition 9.** *A **monoid** is a tuple $(\mathbb{K}, \odot, \boldsymbol{I})$, where $\mathbb{K}$ is a set, $\odot$ is an associative binary operation, and $\boldsymbol{I} \in \mathbb{K}$, called the **identity** element, satisfies $\boldsymbol{I} \odot a = a \odot \boldsymbol{I} = a$ for all $a \in \mathbb{K}$. If $a \odot b = b \odot a$ for all $a, b \in \mathbb{K}$, we say that the monoid is **commutative**.*

**Definition 10.** *A **semiring** is a tuple $(\mathbb{K}, \oplus, \otimes, \boldsymbol{0}, \boldsymbol{1})$ where $(\mathbb{K}, \oplus, \boldsymbol{0})$ is a commutative monoid and $(\mathbb{K}, \otimes, \boldsymbol{1})$ is a monoid. Additionally, $\otimes$ distributes over $\oplus$: $a \otimes (b \oplus c) = (a \otimes b) \oplus (a \otimes c)$ and $(a \oplus b) \otimes c = (a \otimes c) \oplus (b \otimes c)$; furthermore, $\boldsymbol{0}$ is absorbing with respect to $\otimes$: $\boldsymbol{0} \otimes a = a \otimes \boldsymbol{0} = \boldsymbol{0}$.*

**Definition 11.** *A **weighted context-free grammar (WCFG)** over a semiring $(\mathbb{K}, \oplus, \otimes, \boldsymbol{0}, \boldsymbol{1})$ is a tuple $\mathcal{G} = (V, \Sigma, R, S)$ where (1) $V$ is a finite set called the **variables**, (2) $\Sigma$ is a finite set, disjoint from $V$, called the **terminals**, (3) $R \colon V \times (V \cup \Sigma)^* \to \mathbb{K}$ is a **production weighting function**, and (4) $S \in V$ is the **start variable**. If $R(X \to \alpha) = w$, we write $X \xrightarrow{w} \alpha$.*

A WCFG generates strings with some weight, and the weight's semantics are given by the semiring. We can recover the unweighted version of a WCFG using the **Boolean semiring** $(\{0, 1\}, \vee, \wedge, 0, 1)$. A WCFG can assign probabilities to strings with the **real semiring** $(\mathbb{R}_{\geq 0}, +, \times, 0, 1)$ or, for numerical stability, the **log semiring** $(\mathbb{R} \cup \{-\infty\}, \log(\exp(\cdot) + \exp(\cdot)), +, -\infty, 0)$. A WCFG must define a probability distribution in order to allow sampling from it. One way to achieve this is by normalizing the WCFG per variable.

**Definition 12.** *A WCFG $\mathcal{G} = (V, \Sigma, R, S)$ over the real semiring is called a **probabilistic CFG (PCFG)** if for all $X \in V$, $\sum_{X \xrightarrow{p} \alpha} p = 1$.*

To test the ability of neural networks to generalize to lengths unseen during training, we must be able to control the length of the sampled strings for all of the training, validation, and test sets. Generally, one could use rejection sampling (Petty et al., 2025) to exclude strings with lengths that fall outside of the desired length range, but this would be inefficient. Instead, we extend the method proposed by Butoi et al. (2025) to context-free grammars.

Let $\mathcal{G}$ be a *trim* CFG, which we convert to a probabilistic CFG $\mathcal{G}'$ that defines the probability distribution $p_{\mathcal{G}'}$. Additionally, let $N_{\mathcal{G}'} \stackrel{\text{def}}{=} \{n \in [n_{\min}, n_{\max}] \mid \exists \boldsymbol{w} \in \Sigma^n, p_{\mathcal{G}'}(\boldsymbol{w}) > 0\}$ be the set of valid lengths, and W a $p_{\mathcal{G}'}$-distributed random variable. We sample strings from the distribution

$$p(\boldsymbol{w}) = \mathbb{1}[\, |\boldsymbol{w}| \in N_{\mathcal{G}'}] \frac{1}{|N_{\mathcal{G}'}|} p_{\mathcal{G}'}(\mathrm{W} = \boldsymbol{w} \mid |\mathrm{W}| = |\boldsymbol{w}|). \tag{2}$$

by first sampling a length $n$ uniformly at random from $N_{\mathcal{G}'}$, and then by sampling a string $\boldsymbol{w}$ from the conditional distribution $p_{\mathcal{G}'}(\mathrm{W} = \boldsymbol{w} \mid |\mathrm{W}| = n)$. To sample from this distribution, one must filter the derivations of strings $\boldsymbol{w}$ such that $|\boldsymbol{w}| = n$ and renormalize their weights to ensure they form a probability distribution. This renormalization involves computing the **allsum**, i.e., the total weight of all derivations. This is an expensive operation but it can be computed at once for all lengths, using a WCFG defined over a specific semiring, called the binning semiring (Snæbjarnarson et al., 2025).

**Definition 13.** *Let $\mathcal{W} = (\mathbb{K}, \oplus, \otimes, \boldsymbol{0}, \boldsymbol{1})$ be a semiring, and let $D \in \mathbb{Z}_{\geq 0}$. For $\boldsymbol{v} \in \mathbb{K}^{D+1}$, we write $\boldsymbol{v} = (\boldsymbol{v}_0, \boldsymbol{v}_1, \ldots, \boldsymbol{v}_D)$.[5] The $D^{th}$-**order binning semiring** with respect to the **base semiring** $\mathcal{W}$ is the*

---

[5] The vectors $\boldsymbol{v} = (\boldsymbol{v}_0, \boldsymbol{v}_1, \ldots, \boldsymbol{v}_D)$ contain the coefficients of a formal polynomial of the form $\sum_{i=0}^{D} \boldsymbol{v}_i X^i$. The set $\mathbb{K}^{D+1}$ can therefore be seen as a subset of the polynomial ring in $X$ over $\mathbb{K}$, usually denoted as $\mathbb{K}[X]$, for some fixed $D$.

*semiring* $\mathcal{W}^D = (\mathbb{K}^{D+1}, \bigoplus, \bigotimes, \mathbf{0}, \mathbf{1})$, *where:*

$$(\boldsymbol{u} \bigoplus \boldsymbol{v})_i \stackrel{\text{def}}{=} \boldsymbol{u}_i \oplus \boldsymbol{v}_i \qquad\qquad (\boldsymbol{u}, \boldsymbol{v} \in \mathbb{K}^{D+1}; 0 \le i \le D) \tag{3a}$$

$$(\boldsymbol{u} \bigotimes \boldsymbol{v})_i \stackrel{\text{def}}{=} \bigoplus_{j=0}^{i} \boldsymbol{u}_j \otimes \boldsymbol{v}_{i-j} \qquad\qquad (\boldsymbol{u}, \boldsymbol{v} \in \mathbb{K}^{D+1}; 0 \le i \le D) \tag{3b}$$

$$\mathbf{0} \stackrel{\text{def}}{=} (\mathbf{0}, \dots, \mathbf{0}) \qquad\qquad \mathbf{1} \stackrel{\text{def}}{=} (\mathbf{1}, \mathbf{0}, \dots, \mathbf{0}) \tag{3c}$$

The indices of the vectors in the $D^{\text{th}}$-order binning semiring represent the values of an integer counter from 0 to $D$; the meaning of the counter depends on how the semiring is used. We use the counter to keep track of the number of symbols generated by the CFG. To do length-constrained sampling from the PCFG $\mathcal{G}'$, we **lift** it to a $n_{\max}{}^{\text{th}}$-order binning semiring-weighted CFG $\mathcal{G}_D$ as follows. For every rule $X \xrightarrow{p} a$ in $\mathcal{G}'$, we set the weight of $X \to a$ in $\mathcal{G}_D$ to $(\mathbf{0}, p, \mathbf{0}, \dots, \mathbf{0})$, indicating that the rule generates exactly one symbol with probability $p$. We set the weight of rules $X \xrightarrow{p} YZ$ or $X \xrightarrow{p} \varepsilon$ in $\mathcal{G}'$ to $(p, \mathbf{0}, \mathbf{0}, \dots, \mathbf{0})$, indicating that they generate no symbols. A similar interpretation applies to anything with a weight, including derivations and strings.

When computing the allsum of a WCFG over the binning semiring, we get the total weight of all derivations for all lengths $n \in [n_{\min}, n_{\max}]$ at once. This allows us to renormalize the weights of its derivations, in $O(|R| \times D^2)$ time, such that $p_{\mathcal{G}'}(\mathrm{W} = \boldsymbol{w} \mid |\mathrm{W}| = n)$ forms a valid probability distribution for each $n$. Having computed these weights, we can sample strings of some desired length $n$ in linear time. We give additional details about the length-constrained sampling procedure from context-free languages in §4.2.1.

### 4.3 TRAINING AND EVALUATION

We train a neural network to classify whether an input string is in $L$ by minimizing the binary cross-entropy (cf. Weiss et al., 2018; Bhattamishra et al., 2023; van der Poel et al., 2024; Hahn & Rofin, 2024; Butoi et al., 2025). For any probability $p$ and proposition $\phi$, the binary cross-entropy $H_\phi(p)$ is defined as follows:

$$H_\phi(p) \stackrel{\text{def}}{=} \begin{cases} -\log(p) & \text{if } \phi \\ -\log(1-p) & \text{otherwise.} \end{cases} \tag{4}$$

Then for some model $M$ and string $\boldsymbol{w}$ we minimize the loss function

$$\mathcal{L}(M, \boldsymbol{w}) \stackrel{\text{def}}{=} H_{\boldsymbol{w} \in L}(p_M(\boldsymbol{w} \in L)). \tag{5}$$

## 5 EXPERIMENTS

Our experimental setup follows the framework of Butoi et al. (2025), which offers a systematic methodology for evaluating neural networks as formal language recognizers. We compare three neural network architectures that have been widely studied in the formal language learning literature: vanilla RNNs (Elman, 1990), LSTMs (Gers & Schmidhuber, 2001), and causally-masked transformers (Vaswani et al., 2017). See App. E for details of the architectures. For each language, we sample 10k training examples with lengths in the range $[0, 40]$. We further experiment with larger training sets, up to 100k examples, to assess whether additional training data enhances model performance. Following Butoi et al. (2025), we construct two validation sets of 1k examples each: a short set matching the training length range, and a long set with strings in an extended range $[0, 80]$. To assess length generalization, we use test sets containing $5,010$ strings with lengths in the range $[0, 500]$, with approximately 10 strings per length. We also train three non-neural baselines, namely an n-gram model, an SVM, and logistic regression, to get a rough estimate of the difficulty of the languages sampled from each language class. See App. K for details of the baselines. All models are trained as binary classifiers to output whether an input string belongs to the target language.

The evaluation focuses on two aspects: *inductive bias*, measured on the short validation set to determine what the models tend to learn in the absence of guidance on how to generalize, and *expressivity*, measured on the long validation set to assess whether the models can learn the correct

Table 1: Accuracy achieved by transformers, RNNs, and LSTMs on 20 regular (R), partially-ordered DFA (PODFA), star-free (SF), and context-free (CF) languages on test strings in the length range $[0, 500]$. For each language, we train 10 models with different parameter settings. For "Inductive bias", we use validation strings in the length range $[0, 40]$ and report the mean accuracy across all languages. For "Expressivity", we use validation strings in the length range $[0, 80]$ and report the mean across the best-performing trials.

| Language | Inductive Bias | | | Expressivity | | |
|---|---|---|---|---|---|---|
| | Tf | RNN | LSTM | Tf | RNN | LSTM |
| PODFA | $0.99_{\pm 0.01}$ | $0.97_{\pm 0.03}$ | $1.00_{\pm 0.00}$ | $0.99_{\pm 0.00}$ | $0.99_{\pm 0.01}$ | $1.00_{\pm 0.00}$ |
| SF | $0.86_{\pm 0.14}$ | $0.68_{\pm 0.17}$ | $0.95_{\pm 0.06}$ | $0.90_{\pm 0.11}$ | $0.79_{\pm 0.18}$ | $0.99_{\pm 0.04}$ |
| R | $0.63_{\pm 0.04}$ | $0.62_{\pm 0.10}$ | $0.79_{\pm 0.16}$ | $0.66_{\pm 0.04}$ | $0.68_{\pm 0.10}$ | $0.81_{\pm 0.15}$ |
| CF | $0.93_{\pm 0.08}$ | $0.93_{\pm 0.07}$ | $0.95_{\pm 0.06}$ | $0.94_{\pm 0.07}$ | $0.95_{\pm 0.06}$ | $0.96_{\pm 0.05}$ |

Table 2: Percentage of regular (R), partially-ordered DFA (PODFA), star-free (SF), and context-free (CF) languages learned perfectly by transformers, RNNs, and LSTMs. For each architecture and language class, we train 10 models on each of the 20 languages with different hyperparameter settings. For each language class and architecture, we report the number of languages on which at least one of the models achieves perfect accuracy. For "Inductive bias" we use validation strings in the length range $[0, 40]$; for "Expressivity" we use validation strings in the length range $[0, 80]$.

| Language | Inductive Bias | | | Expressivity | | |
|---|---|---|---|---|---|---|
| | Tf | RNN | LSTM | Tf | RNN | LSTM |
| PODFA | 5% | 25% | 70% | 70% | 70% | 70% |
| SF | 15% | 15% | 50% | 50% | 50% | 55% |
| R | 0% | 5% | 35% | 35% | 35% | 35% |
| CF | 5% | 5% | 15% | 15% | 15% | 15% |

algorithm when given hints about extrapolation to longer sequences. For each language class, we generate 20 random languages. Unlike Butoi et al. (2025), we do not constrain models to have the same number of parameters. Instead, for each architecture and language, we train models of varying sizes (approx. 128k, 256k, 512k, and 1024k parameters), and we report the performance of the best model. We also perform a series of experiments in which we fix the number of parameters, to check whether increasing the model size boosts performance. All models use 5 layers, and we automatically adjust the hidden dimension so that the total parameter count is as close as possible to the desired parameter budget. For each experiment, we run 10 trials with different hyperparameter configurations. Specifically, every time we train a model, we randomly sample the batch size and initial learning rate. We use early stopping and learning rate decay. See App. F for further experimental details. For inductive bias experiments, we report the mean performance across all models, while for expressivity experiments, we report the mean over the best-performing trials for each language class.

Finally, we investigate whether structural properties of the target languages correlate with model performance (Valvoda et al., 2022; Borenstein et al., 2024). For automata, we measure accuracy of the best-performing models for each architecture as a function of the number of states and alphabet size; for context-free languages, as a function of alphabet size, number of variables, and number of rules. For star-free languages, we also measure accuracy as a function of dot-depth. We plot the average cross entropy per language class as a function of input length to assess whether performance degrades as the input length increases (Delétang et al., 2023; Butoi et al., 2025). For context-free languages, we also assess whether misclassifications of the negative examples are correlated to the edit distance from the target language.

## 6 RESULTS AND DISCUSSION

We generate 20 random regular, partially-ordered star-free, star-free of varying dot-depth, and context-free languages, and we train 10 models per model size for each architecture, with different hyperparameter settings. We show the mean accuracy per language class in Table 1. We find that the

accuracy can differ significantly between languages from the same language class, and the average accuracy does not tell us whether the models can actually *learn* the languages, i.e., make a correct prediction for all test strings. Therefore, we also show the number of languages learned perfectly by the models (Table 2), to be able to compare our results to those of Butoi et al. (2022). In most cases, the models perform significantly better in the expressivity experiments, indicating that providing them with guidance on how to generalize to longer sequences is crucial. For both transformers and RNNs, however, there is a pronounced gap between their performance in the two types of experiments: in the inductive bias setting, they fail to learn most languages, even those from the relatively simple partially-ordered star-free class.

To estimate the inherent difficulty of the languages sampled from each class, we train three non-neural baselines: an n-gram model, an SVM, and logistic regression (App. K). All baselines achieve very high accuracy on the context-free languages, suggesting that randomly sampled context-free languages tend to be intrinsically easier than those from other classes. In most cases, the neural models outperform the baselines; however, the n-gram model trained on regular languages surpasses both the transformer and the RNN.

Both Transformers and RNNs perform poorly on regular languages. For transformers, this aligns with a long line of theoretical work suggesting that they cannot capture the full class of regular languages. The LSTM, on the other hand, achieves near-perfect accuracy when the dataset size is large enough (App. I). The PODFA languages appear to be the easiest to learn, with all architectures achieving near-perfect accuracy. In contrast, performance drops sharply on star-free languages of varying dot-depths. This finding is consistent with the recent theoretical result of Li & Cotterell (2025), which proves that under more realistic assumptions, transformers can learn only a narrow subset of star-free languages—specifically, those expressible in a restricted fragment of linear temporal logic, equivalent to the languages encoded by PODFAs. All models achieve around 95% average accuracy on context-free languages. However, they master only a small subset of these languages perfectly, suggesting that while the models capture useful approximations, they still fail to learn the underlying generative mechanisms. To test this hypothesis, we plot the cross-entropy of negative strings as a function of edit distance (App. L). We find that most misclassifications cluster at small edit distances, suggesting that the models are relying on superficial cues rather than capturing the true structure of the target language. For comparison, we generate analogous plots for the star-free languages and observe that, in contrast, misclassifications in these languages are not correlated with edit distance.

The size of the machine has a big impact on the accuracy for all architectures: in general, the performance drops when the number of states, alphabet size, dot-depth number of variables or rules increases (see App. G). Moreover, when measuring cross-entropy as a function of input sequence length, we observe that for both regular and star-free languages, cross-entropy rises for strings whose lengths fall outside the training range (see App. H). Finally, we check whether increasing the number of parameters boosts performance. We find that for the transformer and LSTM the performance stays approximately the same, and for the RNN the performance decreases slightly with the size of the model (App. J).

## 7 CONCLUSION

In this work, we systematically evaluate the ability of transformers, RNNs, and LSTMs to recognize randomly sampled formal languages across multiple classes of the Chomsky hierarchy. To support this evaluation, we develop new algorithms for sampling random languages and for length-constrained sampling from context-free languages. By moving beyond a handful of hand-picked languages from each class, our approach provides a more comprehensive assessment of what these models can actually learn in practice. The results reveal a nuanced picture: all architectures succeed on certain subclasses such as partially-ordered star-free languages, yet they struggle with more general star-free languages despite theoretical expressivity guarantees. For context-free languages, their seemingly strong performance is misleading: we show that none of the architectures actually learns the correct underlying mechanisms. These findings underscore the gap between theoretical analyses and empirical learnability, and they suggest that formal language recognition remains a powerful tool for probing the algorithmic capabilities and limitations of neural network architectures.

ETHICS STATEMENT

Our work is primarily theoretical, focusing on the development of new algorithms and the evaluation of neural network architectures on formal languages. All experiments are conducted on artificially generated data rather than real-world or sensitive datasets. As such, we do not anticipate any direct ethical concerns arising from this research. We used large language models solely to aid and polish writing.

REPRODUCIBILITY STATEMENT

The supplementary code submitted with the paper is self-contained and includes the necessary information to generate the data and reproduce our experiments. Additionally, the appendix contains further details about the algorithms used in this paper. App. B contains additional details about our language sampling algorithms, App. C restates details about length-constrained sampling from regular languages, and App. D provides additional information about our length-constrained sampling algorithm from context-free languages.

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

## A  FURTHER BACKGROUND

### A.1  SEMIRINGS

A key component of our method is computing the allsum over some particular semiring. Computing the allsum requires summing over an infinite number of derivations of a CFG, which requires the ability to perform infinite summations in its semiring.

**Definition 14.** *Let $(\mathbb{K}, \oplus, \otimes, \mathbf{0}, \mathbf{1})$ be a semiring. Let $a^{\otimes i} = \bigotimes_{j=1}^{i} a$, and $a^* = \bigoplus_{i=0}^{\infty} a^{\otimes i}$. If $a^*$ is defined and in $\mathbb{K}$ for all $a \in \mathbb{K}$, we say the semiring is **closed**.*

In the real semiring, $a^* = \frac{1}{1-a}$. In the log semiring, $a^* = \log \frac{1}{1-\exp a} = -\log(1 - \exp(a))$.

If a semiring $\mathcal{W} = (\mathbb{K}, \oplus, \otimes, \mathbf{0}, \mathbf{1})$ is closed, so is any $D^{\text{th}}$-order binning semiring with respect to $\mathcal{W}$, i.e., $\mathcal{W}^D = (\mathbb{K}^{D+1}, \oplus, \otimes, \mathbf{0}, \mathbf{1})$. We write $\boldsymbol{v}^{\circledast} \stackrel{\text{def}}{=} \oplus_{j=0}^{\infty} \boldsymbol{v}^{\otimes j}$. It has the following closed-form solution.

$$(\boldsymbol{v}^{\circledast})_i = \boldsymbol{v}_0^* \otimes \left( \mathbf{1}_i \oplus \bigoplus_{j=1}^{i} \boldsymbol{v}_j \otimes (\boldsymbol{v}^{\circledast})_{i-j} \right) \qquad (0 \le i \le D) \qquad (6)$$

A derivation is given by Snæbjarnarson et al. (2025). The elements $(\boldsymbol{v}^{\circledast})_i$ can be computed in order of increasing $i$. Assuming $\oplus$, $\otimes$, and $^*$ run in $O(1)$ time, the total time complexity is $O(D^2)$, since it involves $O(D)$ iterations, each of which takes $O(D)$ time.

## A.2 FINITE AUTOMATA

**Definition 15.** *A **path** $\pi$ of a DFA $\mathcal{A} = (Q, \Sigma, \delta, q_0, F)$ is a sequence of connecting states and transitions,*

$$\pi = r_0 \xrightarrow{w_1} r_1 \xrightarrow{w_2} r_2 \cdots r_{m-1} \xrightarrow{w_n} r_m, \qquad (7)$$

*where $r_i \xrightarrow{w_{i+1}} r_{i+1} \in \delta$ for all $i = 0, \ldots, m-1$. If $\pi$'s transitions scan the sequence of symbols $w_1, \ldots, w_n$, we say that $\mathcal{A}$ **scans** the string $\boldsymbol{w} = w_1 \cdots w_n$.*

A DFA accepts a string $\boldsymbol{w}$ if it has a path from the initial state $q_0$ to a final state $r \in F$ that scans $\boldsymbol{w}$. Importantly, a DFA has at most one path scanning $\boldsymbol{w}$ for every string $\boldsymbol{w} \in \Sigma^*$. This allows a linear-time algorithm for checking whether a string is accepted by a DFA (see Algorithm 1).

---

**Algorithm 1** Check whether a DFA $\mathcal{A}$ accepts a string $\boldsymbol{w} \in \Sigma^*$. Time complexity: $O(n)$.

---
1. **def** SIMULATEDFA($\mathcal{A} = (Q, \Sigma, \delta, q_0, F), \boldsymbol{w} = w_1 \cdots w_n$):
2.    $q \leftarrow q_0$
3.    **for** $a = w_1, \ldots, w_n$ :
4.      $q \leftarrow \delta(q, a)$
5.      **if** $q = \emptyset$ :
6.        **return** $\perp$
7.    **return** $q \in F$

---

**Definition 16.** *Let $\mathcal{A} = (Q, \Sigma, \delta, q_0, F)$ be a DFA/NFA. We call a state $q \in Q$ **accessible** if $\mathcal{A}$ has a path from $q_0$ to $q$ and **co-accessible** if $\mathcal{A}$ has a path from $q$ to $r$, where $r \in F$. We call $\mathcal{A}$ **trim** if all its states are both accessible and co-accessible.*

We call a state **useless** if it is either non-accessible or non-co-accessible. We call a transition **useless** if it contains any useless states. Importantly, the correctness of the length-constrained sampling method for regular languages relies on the fact that the DFA is trim. We outline how to remove useless states and transitions from a DFA in Algorithm 2.

A DFA can be weighted with weights from a semiring, resulting in a device that assigns weights to strings in $\Sigma^*$ rather than just accepting or rejecting them.

**Definition 17.** *A **weighted finite automaton (WDFA)** over semiring $(\mathbb{K}, \oplus, \otimes, \mathbf{0}, \mathbf{1})$ is a tuple $\mathcal{A} = (Q, \Sigma, \delta, q_0, \rho)$ such that (1) $Q$, $\Sigma$, $q_0$, and $F$ are defined as in Def. 1; and (2) $\delta \colon Q \times (\Sigma \cup \{\varepsilon\}) \times Q \to \mathbb{K}$ is the **transition function**. If $\delta(q, a, r) = w \ne \mathbf{0}$, we say that $\mathcal{A}$ has a transition from $q$ to $r$ that scans $a$ with weight $w$, and we write $q \xrightarrow{a/w} r \in \delta$.*

The paths of a WDFA are sequences of connecting states and *weighted* transitions.

**Definition 18.** *Let $\mathcal{A} = (Q, \Sigma, \delta, q_0, \rho)$ be a WDFA. The **inner weight** of a path $\pi = r_0 \xrightarrow{a_1/w_1} r_1 \xrightarrow{a_2/w_2} r_2 \cdots r_{m-1} \xrightarrow{a_m/w_m} r_m$ is defined as*

$$\mathbf{w}_{\mathrm{I}}(\pi) \stackrel{\text{def}}{=} \bigotimes_{i=1}^{m} w_i. \qquad (8)$$

---

**Algorithm 2** Trim useless states and transitions from an DFA/NFA $\mathcal{A}$. The algorithm REACHABLE$(G = (V, E), S)$ can be any graph search algorithm that returns the set of all vertices in $V$ that are reachable from any vertex in $S$ in $O(|V| + |E|)$ time.

1. **def** TRIMNFA$(\mathcal{A} = (Q, \Sigma, \delta, q_0, F))$:
2. $\quad G \leftarrow (Q, \{(q, r) \colon q \xrightarrow{a} r \in \delta\})$
3. $\quad Q' \leftarrow$ REACHABLE$(G, \{q_0\})$
4. $\quad G \leftarrow (Q', \{(r, q) \colon q \xrightarrow{a} r \in \delta \mid q, r \in Q'\})$
5. $\quad F' \leftarrow F \cap Q'$
6. $\quad Q' \leftarrow$ REACHABLE$(G, F')$
7. $\quad \delta' \leftarrow \{q \xrightarrow{a} r \in \delta \mid q, r \in Q'\}$
8. $\quad \Sigma' \leftarrow \{a \colon q \xrightarrow{a} r \in \delta'\}$
9. $\quad$ add $q_0$ to $Q'$
10. $\quad \mathcal{A}' \leftarrow (Q', \Sigma', \delta', q_0, F')$
11. $\quad$ **return** $\mathcal{A}'$

---

*The **path weight** of $\pi$ is defined as*

$$\mathbf{w}(\pi) \stackrel{\text{def}}{=} \mathbf{w}_{\mathrm{I}}(\pi) \otimes \rho(r_m). \tag{9}$$

The inner path weight can be computed efficiently for all pairs of states using Lehmann's algorithm (Lehmann, 1977).[6]

Let $\Pi(\mathcal{A}, q \rightsquigarrow r)$ be the set of all paths of $\mathcal{A}$ from $q$ to $r$. The algorithm takes as input a table $A$ defined as

$$A[q, r] \stackrel{\text{def}}{=} \bigoplus_{q \xrightarrow{a/w} r} w \tag{10}$$

for all $q, r \in Q$ and it produces another table $A'[q, r]$ defined as

$$A'[q, r] \stackrel{\text{def}}{=} \bigoplus_{\pi \in \Pi(\mathcal{A}, q \rightsquigarrow r)} \mathbf{w}_{\mathrm{I}}(\pi) \tag{11}$$

for all $q, r \in Q$. We show this procedure in Algorithm 3.

---

**Algorithm 3** Lehmann's algorithm for inverting matrix $A \in \mathbb{K}^{N \times N}$ in the closed semiring $(\mathbb{K}, \oplus, \otimes, \mathbf{0}, \mathbf{1})$.

1. **def** LEHMANN$(A)$:
2. $\quad$ **for** $k = 1, \ldots, N$ :
3. $\quad\quad$ let $A'$ be a $N \times N$ matrix full of $\mathbf{0}$
4. $\quad\quad a \leftarrow A[k, k]^*$
5. $\quad\quad$ **for** $i = 1, \ldots, N$ :
6. $\quad\quad\quad$ **for** $j = 1, \ldots, N$ :
7. $\quad\quad\quad\quad A'[i, j] \leftarrow A[i, j] \oplus (A[i, k] \otimes a \otimes A[k, j])$
8. $\quad\quad A \leftarrow A'$
9. $\quad$ **for** $k = 1, \ldots, N$ :
10. $\quad\quad A[k, k] \leftarrow A[k, k] \oplus \mathbf{1}$
11. $\quad$ **return** $A$

---

**Definition 19.** *Let $\mathcal{A}$ be a WDFA. The **backward weight** $\boldsymbol{\beta}_{\mathcal{A}}(q)$ is the total weight of all paths from $q$ to any other state,*

$$\boldsymbol{\beta}_{\mathcal{A}}(q) \stackrel{\text{def}}{=} \bigoplus_{\substack{r \in Q, \\ \pi \in \Pi(\mathcal{A}, q \rightsquigarrow r)}} \mathbf{w}_{\mathrm{I}}(\pi). \tag{12}$$

The quantity $\boldsymbol{z} = \boldsymbol{\beta}_{\mathcal{A}}(q_0)$ is called the **allsum** and it represents the total weight of all paths of a WDFA $\mathcal{A}$. Algorithm 4 shows how to compute the backward weights of all states of a WDFA $\mathcal{A}$.

---

[6]Lehmann's algorithm requires a WDFA defined over a closed semiring.

---

**Algorithm 4** Backward algorithm on a WDFA $\mathcal{A}$ over the closed semiring $(\mathbb{K}, \oplus, \otimes, \mathbf{0}, \mathbf{1})$.

1. **def** BACKWARD($\mathcal{A} = (Q, \Sigma, \delta, q_0, \rho)$):
2.    let $A$ be a matrix indexed by $Q \times Q$ full of $\mathbf{0}$
3.    **for** $q \xrightarrow{a/w} r \in \delta$ :
4.      $A[q, r] \leftarrow A[q, r] \oplus w$
5.    $A \leftarrow$ LEHMANN($A$)
6.    let $\boldsymbol{\beta}$ be a table indexed by $Q$
7.    **for** $q \in Q$ :
8.      $\boldsymbol{\beta}[q] \leftarrow \bigoplus_{r \in Q} A[q, r] \otimes \rho(r)$
9.    **return** $\boldsymbol{\beta}$

---

**Definition 20.** *A WDFA $\mathcal{A} = (Q, \Sigma, \delta, q_0, \rho)$ over the real semiring is called a **probabilistic finite automaton (PFA)** if (1) for all $q \in Q$, $\left( \sum_{q \xrightarrow{a/p} r \in \delta} p \right) + \rho(q) = 1$, and (2) $\mathcal{A}$ induces a probability distribution over $\Sigma^*$.*

Note that the condition in item (2) is known as tightness and is easily checkable for WDFAs (Du et al., 2023).

### A.3 CONTEXT-FREE GRAMMARS

**Definition 21.** *A variable $X \in V$ is called **accessible** if there exists a derivation $S \overset{*}{\Rightarrow} \alpha_1 X \alpha_2$, where $\alpha_1, \alpha_2 \in (V \cup \Sigma)^*$.*

**Definition 22.** *A variable $X \in V$ is called **co-accessible** if there exists a derivation $X \overset{*}{\Rightarrow} \boldsymbol{w}$ where $\boldsymbol{w} \in \Sigma^*$.*

**Definition 23.** *A CFG is called **trim** if all its variables are both accessible and co-accessible.*

We call a variable **useless** if it is either non-accessible or non-co-accessible. Similarly, we call a rule **useless** if it contains useless variables. Algorithm 5 shows how to remove useless variables and rules from a CFG in Chomsky normal form.

---

**Algorithm 5** Trim useless variables and rules from a CFG $\mathcal{G}$ in Chomsky normal form. The algorithm REACHABLE($G = (V, E), S$) can be any graph search algorithm that returns the set of all vertices in $V$ that are reachable from any vertex in $S$ in $O(|V| + |E|)$ time. Lines 2–6 are equivalent to running weight pushing in the Boolean semiring. Time complexity: $O(|V|^2 + |V||R| + |R|)$.

1. **def** TRIMCFG($\mathcal{G} = (V, \Sigma, R, S)$):
2.    $V' \leftarrow$ NONEMPTYVARIABLES($\mathcal{G}$)
3.    $E \leftarrow \emptyset$
4.    **for** $X \to YZ \in R$ :
5.      **if** $Y \in V'$ and $Z \in V'$ :
6.        add $(X, Y)$ and $(X, Z)$ to $E$
7.    $V' \leftarrow$ REACHABLE($(V' \cup \{S\}, E), \{S\}$)
8.    $R' \leftarrow \{S \to \varepsilon \in R\} \cup \{X \to YZ \in R \mid X, Y, Z \in V'\} \cup \{X \to a \in R \mid X \in V'\}$
9.    $\Sigma' \leftarrow \{a \colon X \to a \in R'\}$
10.   $\mathcal{G}' \leftarrow (V', \Sigma', R', S)$
11.   **return** $\mathcal{G}'$

---

**Definition 24.** *Let $\mathcal{G} = (V, \Sigma, R, S)$ be a WCFG over the semiring $(\mathbb{K}, \oplus, \otimes, \mathbf{0}, \mathbf{1})$. A **partial derivation** $d$ is a sequence of sentential forms and rules,*

$$d = \alpha_0 \overset{r_1}{\Longrightarrow} \alpha_1 \overset{r_2}{\Longrightarrow} \alpha_2 \cdots \alpha_{m-1} \overset{r_m}{\Longrightarrow} \alpha_m, \tag{13}$$

---

**Algorithm 6** Given a CFG $\mathcal{G}$ in Chomsky normal form, determine the set of variables in $V \setminus \{S\}$ that generate at least one string. Time complexity: $O(|V|^2 + |V||R| + |R|)$.

---

1. **def** NONEMPTYVARIABLES($\mathcal{G} = (V, \Sigma, R, S)$):
2.   $R'_{\text{bin}} \leftarrow \{X \rightarrow YZ \in R \mid X \neq S\}$
3.   $T_0 \leftarrow \{X : X \rightarrow a \in R \mid X \neq S\}$
4.   $T \leftarrow T_0$
5.   **loop**
6.     $T' \leftarrow T_0$
7.     **for** $X \rightarrow YZ \in R'_{\text{bin}}$ :
8.       **if** $Y \in T$ and $Z \in T$ :
9.         add $X$ to $T'$
10.     **if** $|T'| = |T|$ :
11.       **return** $T$
12.     $T \leftarrow T'$

---

such that $\alpha_0 = X \in V$, $\alpha_m = \boldsymbol{w} \in \Sigma^*$, and for all $i = 1, \ldots, m$, $r_i \in R$ and $\alpha_{i-1} \xRightarrow{r_i} \alpha_i$. If $\alpha_0 = S$, $d$ is called a **derivation**. The **weight** of $d$ is defined as

$$\mathbf{w}(d) \overset{\text{def}}{=} \bigotimes_{i=1}^{m} R(r_i). \tag{14}$$

**Definition 25.** *Let $\Delta(\mathcal{G}, \boldsymbol{w})$ be the set of all derivations of $\mathcal{G}$ of the string $\boldsymbol{w}$. The **weight** $\mathcal{G}(\boldsymbol{w})$ of a string $\boldsymbol{w}$ is the total weight of all derivations of $\boldsymbol{w}$ of $\mathcal{G}$,*

$$\mathcal{G}(\boldsymbol{w}) \overset{\text{def}}{=} \bigoplus_{d \in \Delta(\mathcal{G}, \boldsymbol{w})} \mathbf{w}(d). \tag{15}$$

A CFG can have multiple derivations of a string $\boldsymbol{w}$, therefore the string weight must account for the weights of all these derivations. When the CFG is in CNF, this quantity can be computed efficiently using the CKY algorithm (Cocke, 1969; Kasami, 1965; Younger, 1967). The runtime of CKY is $O(n^3 |V|^3)$ time, where $n$ is the length of the string. We show in Algorithm 7 the CKY algorithm in the Boolean semiring, which can be used for checking whether a CFG generates a string.

---

**Algorithm 7** Cocke–Kasami–Younger (CKY) Algorithm. Check whether a CNF CFG $\mathcal{G}$ generates a string $\boldsymbol{w} \in \Sigma^*$. Time complexity: $O(|V|^3 n^3)$.

---

1. **def** CKY($\mathcal{G} = (V, \Sigma, R, S), \boldsymbol{w} = w_1 \cdots w_n$):
2.   let $T$ be a table mapping $i, j$ where $1 \leq i \leq j \leq n$ to subsets of $V$
3.   **for** $j = 1, \ldots, n$ :
4.     $T[j, j] \leftarrow \{X : X \rightarrow w_j \in R\}$
5.     **for** $\ell = 2, \ldots, j$ :
6.       $i \leftarrow j - \ell + 1$
7.       **for** $k = i, \ldots, j - 1$ :
8.         **for** $X \rightarrow YZ \in R$ :
9.           **if** $Y \in T[i, k]$ and $Z \in T[k+1, j]$ :
10.             add $X$ to $T[i, j]$
11.   **return** $S \in T[1, n]$

---

# B   DETAILS OF LANGUAGE SAMPLING

Let $\mathcal{U}(A)$ denote the uniform distribution over set $A$, let $\text{Ber}(p)$ denote a Bernoulli distribution with success rate $p$, and let $\text{NB}(r, p)$ denote a negative binomial distribution with $r$ successes and a success rate of $p$.

## B.1 SAMPLING REGULAR LANGUAGES

We randomly sample DFAs with number of states $|Q|$ and alphabet size $|\Sigma|$, where $|Q|$ and $|\Sigma|$ are drawn randomly from negative binomial distributions with means $\mu_Q$ and $\mu_\Sigma$, respectively. Then, we add an outgoing transition with probability $p_\delta$ for each pair $(q, a) \in Q \times \Sigma$ and we sample uniformly its destination state. We show this full procedure in Algorithm 8.

---

**Algorithm 8** Algorithm for randomly sampling a DFA. Here, $\mu_Q \geq 1$ is the mean number of states, $\mu_\Sigma \geq 1$ is the mean alphabet size, $p_\delta \in (0, 1)$ is the probability of adding a transition for a given $(q, a) \in Q \times \Sigma$, and $p_F \in (0, 1)$ is the probability of making a state an accept state. The output is a trim DFA $\mathcal{A}$.

1. **def** SAMPLEDFA($\mu_Q, \mu_\Sigma, p_\delta, p_F$):
2.     sample $|Q| \sim \text{NB}(\mu_Q - 1, 0.5) + 1$
3.     sample $|\Sigma| \sim \text{NB}(\mu_\Sigma - 1, 0.5) + 1$
4.     let $Q = \{q_0, q_1, \ldots, q_{|Q|-1}\}$
5.     let $\Sigma = \{a_1, \ldots, a_{|\Sigma|}\}$
6.     let $\mathcal{A} = (Q, \Sigma, \delta, q_0, F)$ be a DFA
7.     **for** $q \in Q$ :
8.       **for** $a \in \Sigma$ :
9.         **if** $\phi \sim \text{Ber}(p_\delta)$ :
10.           sample $r \sim \mathcal{U}(Q)$
11.           add $q \xrightarrow{a} r$ to $\delta$
12.       **if** $\phi \sim \text{Ber}(p_F)$ :
13.         add $q$ to $F$
14.     $\mathcal{A} \leftarrow \text{TRIMDFA}(\mathcal{A})$
15.     **return** $\mathcal{A}$

---

When sampling random DFAs, some of them might actually accept languages from a lower subclass, such as the star-free languages. It is easy to check whether a DFA is in fact a PODFA by looking at the strongly connected components of the underlying directed graph, i.e., each component contains a single vertex in the case of PODFAs. After we sample a random DFA, we perform this check and we reject any DFA that is in fact a PODFA.

## B.2 SAMPLING LANGUAGES DEFINED BY PODFAS

We randomly sample PODFAs with number of states $|Q|$ and alphabet size $|\Sigma|$, where $|Q|$ and $|\Sigma|$ are drawn randomly from negative binomial distributions with means $\mu_Q$ and $\mu_\Sigma$, respectively. We first assume an ordering $q_0, q_1, \ldots, q_{|Q|-1}$ of the states. Then, we add a transition with probability $p_\delta$ for each pair $(q, a) \in Q \times \Sigma$ and we sample uniformly its destination state from the set of states that come after $q$ in the ordering. We show this procedure in Algorithm 9.

## B.3 SAMPLING STAR-FREE LANGUAGES

To sample star-free languages, we use a generalization of the DFA, namely the nondeterministic finite automaton (NFA). Unlike a DFA, an NFA can reach multiple states upon reading some symbol $a \in \Sigma$ from some state $q \in Q$. Any NFA can be converted into an equivalent DFA through determinization (Hopcroft et al., 2006), but using NFAs makes some of our constructions easier.

**Definition 26.** *A **nondeterministic finite automaton (NFA)** is a tuple $\mathcal{A} = (Q, \Sigma, \delta, q_0, F)$ where (1) $Q$ is a finite set of **states**, (2) $\Sigma$ is an alphabet, (3) $\delta \subseteq Q \times \Sigma \times Q$ is the **transition set**, (4) $q_0 \in Q$ is the **start state**, and (5) $F \subseteq Q$ is the set of **accept states**. If $(q, a, r) \in \delta$, we say that $\mathcal{A}$ has a transition from $q$ to $r$ that scans $a$, and we write $q \xrightarrow{a} r \in \delta$.*

Notions such as paths, accessible and co-accessible states are defined identically as for DFAs. An NFA can have multiple paths scanning some string $w$.

Algorithm 10 samples a DFA with dot-depth upper bounded by some constant $K$. The key idea is to generate automata for each dot-depth $1, \ldots, K$ by alternating between applying concatenation and Boolean operations, starting from the automata encoding the basic languages. We first sample the alphabet size $|\Sigma|$, dot-depth upper bound $K$, number of concatenation operations $m$, and the

---

**Algorithm 9** Algorithm for randomly sampling a PODFA. Here, $\mu_Q \geq 1$ is the mean number of states, $\mu_\Sigma \geq 1$ is the mean alphabet size, $p_\delta \in (0, 1)$ is the probability of adding a transition for a given $(q, a) \in Q \times \Sigma$, and $p_F \in (0, 1)$ is the probability of making a state an accept state. The output is a trim PODFA $\mathcal{A}$.

1. **def** SAMPLEPODFA($\mu_Q, \mu_\Sigma, p_\delta, p_F, p_l$):
2.   sample $|Q| \sim \text{NB}(\mu_Q - 1, 0.5) + 1$
3.   sample $|\Sigma| \sim \text{NB}(\mu_\Sigma - 1, 0.5) + 1$
4.   let $Q = \{q_0, q_1, \ldots, q_{|Q|-1}\}$
5.   let $\Sigma = \{a_1, \ldots, a_{|\Sigma|}\}$
6.   let $\mathcal{A} = (Q, \Sigma, \delta, q_0, F)$ be a DFA
7.   **for** $q_i = q_0, \ldots, q_{|Q|-1}$ :
8.     **for** $a \in \Sigma$ :
9.       **if** $\phi \sim \text{Ber}(p_\delta)$ :
10.         sample $r \sim \mathcal{U}(\{q_i, \ldots, q_{|Q|-1}\})$
11.         add $q_i \xrightarrow{a} r$ to $\delta$
12.     **if** $\phi \sim \text{Ber}(p_F)$ :
13.       add $q_i$ to $F$
14.   $\mathcal{A} \leftarrow$ TRIMDFA($\mathcal{A}$)
15.   **return** $\mathcal{A}$

---

number of Boolean operations $b$ from negative binomial distributions with means $\mu_\Sigma$, $\mu_K$, $\mu_m$, and $\mu_b$, respectively. This step ensures full support over the star-free languages. We construct the automata $\mathcal{A}_\emptyset$, $\mathcal{A}_\varepsilon$, and $\mathcal{A}_a$ for all $a \in \Sigma$ encoding the basic languages for alphabet $\Sigma$. We then generate recursively sets of automata of dot-depths $i = 1, \ldots, K$ by applying random concatenation operations (Algorithm 11) and Boolean operations (Algorithm 12) to automata of dot-depth $i - 1$. For efficiency reasons, we optimize (Algorithm 13) the newly constructed automata by applying determinization, minimization, and trimming (Hopcroft et al., 2006). This process generates sets of automata for each dot-depth $1, \ldots, K$, and we return a single one, uniformly sampled from the set of automata of dot-depth $K$.

## B.4 SAMPLING CONTEXT-FREE LANGUAGES

We sample CFGs in Chomsky normal form with $|V|$ variables, $|\Sigma|$ terminals, $|R_b|$ binary rules, $|R_l|$ lexical rules, and $c$ self-embedding chains, where these numbers are sampled from negative binomial distributions with means $\mu_V$, $\mu_\Sigma$, $\mu_{R_b}$, $\mu_{R_l}$, and $\mu_c$, respectively. This step ensures full support over the context-free languages. Then, we compute the set of all possible rules, and we sample without replacement $|R_b|$ binary rules and $|R_l|$ lexical rules. We trim the resulting CFG, and, if it has more than one variable, we add $c$ self-embedding chains (Algorithm 15). For each chain, its length $k$ is sampled from a negative binomial distribution with mean $\mu_k$. Recall that a self-embedding chain is a partial derivation of the form $X \overset{*}{\Rightarrow} \alpha_1 X \alpha_2$, where $\alpha_1, \alpha_2 \in V^*$. We sample uniformly such a variable $X$, and then sample recursively conforming binary rules. Because the CFG was trim before adding the chains, and because we do not introduce any new variables, the resulting CFG is also trim. This requirement is essential for the length-constrained sampling algorithm.

Sometimes, either because of trimming, or simply because of the combination of rules that are sampled, the resulting CFG is degenerate. For particular combinations of means $\mu_V$, $\mu_\Sigma$, $\mu_{R_b}$, $\mu_{R_l}$, $\mu_c$, and $\mu_k$, we noticed that many sampled CFGs actually encode finite languages, or have a low density of either positive or negative examples, making it difficult to sample strings from. For this reason, we ran a grid search over the hyperparameters $\mu_V$, $\mu_\Sigma$, $\mu_{R_b}$, $\mu_{R_l}$, $\mu_c$, and $\mu_k$, and selected those that gave the lowest percentage of degenerate CFGs. However, we didn't find any configuration where this percentage was exactly 0 in our experiments. Therefore, during sampling, we reject any such CFGs. To check whether a CFG is finite, we look for cycles in the reachability graph (Clark, 2017).

**Algorithm 10** Algorithm for randomly sampling a NFA that encodes a star-free languages of dot-depth bounded by $K$. Here, $\mu_\Sigma$ is the mean number of symbols, $\mu_K \geq 1$ is the mean dot depth, $\mu_m \geq 1$ is the mean number of concatenation operations, $\mu_b \geq 1$ is the mean number of boolean operations, and $p_F$ is the probability of a state being an accept state. The output is a trim DFA $\mathcal{A}$.

1. **def** SAMPLEDOTDEPTH($\mu_K$):
2.    sample $|\Sigma| \sim \mathrm{NB}(\mu_\Sigma - 1, 0.5) + 1$        ▷*Sample alphabet size.*
3.    sample $K \sim \mathrm{NB}(\mu_K - 1, 0.5) + 1$        ▷*Sample dot-depth.*
4.    sample $m \sim \mathrm{NB}(\mu_m - 1, 0.5) + 1$      ▷*Sample number of concatenation operations.*
5.    sample $b \sim \mathrm{NB}(\mu_b - 1, 0.5) + 1$      ▷*Sample number of boolean operations.*
6.    let $\Sigma = \{a_1, \ldots, a_{|\Sigma|}\}$
7.    let $\mathcal{A}_\varepsilon = (\{q_0, q_1\}, \Sigma, \delta_\varepsilon, q_0, \{q_0\})$ and $\delta_\varepsilon = \{q_0 \xrightarrow{a} q_1 \mid a \in \Sigma\} \cup \{q_1 \xrightarrow{a} q_1 \mid a \in \Sigma\}$   ▷*Automaton accepting only $\varepsilon$.*
8.    let $\mathcal{A}_\emptyset = (\{q_0\}, \Sigma, \delta_\emptyset, q_0, \emptyset)$ and $\delta_\emptyset = \{q_0 \xrightarrow{a} q_0 \mid a \in \Sigma\}$     ▷*Automaton encoding $\emptyset$.*
9.    **for** $a \in \Sigma$ :            ▷*Automaton accepting only $a$.*
10.      let $\mathcal{A}_a = (\{q_0, q_1, q_2\}, \Sigma, \delta_a, q_0, \{q_1\})$ and $\delta_a = \{q_0 \xrightarrow{a} q_1\} \cup \{q_0 \xrightarrow{c} q_2 \mid c \in \Sigma, c \neq a\}$
11.    automata $= \{\mathcal{A}_\varepsilon, \mathcal{A}_\emptyset\} \cup \{\mathcal{A}_a \mid a \in \Sigma\}$      ▷*Automata encoding basic languages.*
12.    automataByLevel $=$ [automata]      ▷*List of sets of automata for each level $1 \ldots K$.*
13.    **for** $l \in 1 \ldots K - 1$ :
14.      automata $=$ M(automataByLevel$[l-1], m$)
15.      automata $=$ B(automata, $b, \Sigma$)
16.      append automata to automataByLevel.
17.    sample $\mathcal{A} = (Q, \Sigma, \delta, q_0, F) \sim \mathcal{U}($automataByLevel$[K])$
18.    let $F = \emptyset$
19.    **for** $q \in Q$ :
20.      **if** $\phi \sim \mathrm{Ber}(p_F)$ :
21.        add $q$ to $F$
22.    $\mathcal{A} \leftarrow$ OPTIMIZE($\mathcal{A}$)
23.    **return** $\mathcal{A}$

---

**Algorithm 11** Returns a set of DFAs resulting from applying random concatenation operations to automata from the input set. The algorithm CONCATENATE is a standard NFA concatenation algorithm.

1. **def** M(automata, $m$):
2.    $C = \emptyset$
3.    **for** $i = 1 \ldots m$ :
4.      sample $\mathcal{A}_1 \sim \mathcal{U}($automata$)$
5.      sample $\mathcal{A}_2 \sim \mathcal{U}($automata$)$
6.      $r =$ CONCATENATE($\mathcal{A}_1, \mathcal{A}_2$)
7.      $r =$ OPTIMIZE($r$)
8.      add $r$ to $C$
9.    **return** $C$

---

## C   DETAILS OF LENGTH-CONSTRAINED SAMPLING FOR REGULAR LANGUAGES

We summarize the length-constrained procedure for regular languages used by Butoi et al. (2025).

Let $\mathcal{A}$ be a fixed *trim* DFA and $[n_{\min}, n_{\max}]$ a length range. The following preprocessing steps are run once:

1. Convert $\mathcal{A}$ to a PDFA $\mathcal{A}'$, then convert $\mathcal{A}'$ to a WDFA $\mathcal{A}_D$ over the $n_{\max}$th-order binning semiring with respect to the log semiring (Algorithm 16). Importantly, $\mathcal{A}'$ is tight because $\mathcal{A}$ is trim and the way the probabilities are assigned.

2. Push the weights of $\mathcal{A}_D$ to get a new WDFA $\mathcal{A}'_D$ over the $n_{\max}$th-order binning semiring with respect to the real semiring (Algorithm 17) using the backward algorithm (Algorithm 4) and Lehmann's algorithm (Algorithm 3);

3. Compute the subset of valid lengths $N \subseteq [n_{\min}, n_{\max}]$ (Algorithm 18).

Afterwards, the following steps have to be run to sample one string:

---

**Algorithm 12** Returns a set of DFAs resulting from applying random Boolean operations to automata from the input set. The algorithms UNION, INTERSECT, and COMPLEMENT are standard NFA union, intersection, and complementation algorithms.

---
1. **def** B(automata, $b$):
2.   $B = \emptyset$
3.   Ops $= \{$UNION, INTERSECT, COMPLEMENT$\}$          ▷*Set of Boolean operations.*
4.   **for** $i = 1 \ldots m$ :
5.     sample $\mathcal{A}_1 \sim \mathcal{U}($automata$)$
6.     sample $\mathcal{A}_2 \sim \mathcal{U}($automata$)$
7.     sample OP $\sim \mathcal{U}($Ops$)$
8.     **if** OP $=$ COMPLEMENT :
9.       $r = $ COMPLEMENT$(\mathcal{A}_1)$
10.     **else**
11.       $r = $ OP$(\mathcal{A}_1, \mathcal{A}_2)$
12.     $r = $ OPTIMIZE$(r)$
13.     add $r$ to $C$
14.   **return** $B$

---

**Algorithm 13** Returns an optimized DFA by applying determinization, minimization, and trimming to the input NFA.

---
1. **def** OPTIMIZE$(\mathcal{A})$:
2.   $\mathcal{A} = $ DETERMINIZE$(\mathcal{A})$
3.   $\mathcal{A} = $ MINIMIZE$(\mathcal{A})$
4.   $\mathcal{A} = $ TRIM$(\mathcal{A})$
5.   **return** $\mathcal{A}$

---

1. uniformly sample a length $n$ from $N$;

2. sample a string of length $n$ (Algorithm 19).

# D   DETAILS OF LENGTH-CONSTRAINED SAMPLING FOR CONTEXT-FREE LANGUAGES

Let $\mathcal{G}$ be a fixed context-free grammar in Chomsky normal form, and let $[n_{\min}, n_{\max}]$ denote the length range of interest. To enable string sampling, $\mathcal{G}$ must be endowed with a probability distribution over $\Sigma^*$. A natural way to achieve this is to transform $\mathcal{G}$ into a probabilistic CFG $\mathcal{G}'$ by assigning uniform weights to all rules sharing the same left-hand side variable. The resulting grammar $\mathcal{G}'$ defines a distribution $p_{\mathcal{G}'}(\boldsymbol{w})$ over strings, and sampling from it amounts to sampling *derivations* of $\mathcal{G}'$.

For a fixed length $n$, we seek to sample strings $\boldsymbol{w} \in \Sigma^*$ such that $|\boldsymbol{w}| = n \in [n_{\min}, n_{\max}]$. Equivalently, we aim to sample from the conditional distribution $p_{\mathcal{G}'}(\boldsymbol{w} \mid |\boldsymbol{w}| = n)$. This can be done by considering all derivations of $\mathcal{G}'$ of the form $S \overset{*}{\Rightarrow} \boldsymbol{w}$ with $|\boldsymbol{w}| = n$, and then renormalizing their weights to obtain a proper probability distribution. The renormalization relies on computing a quantity known as the **allsum**.

**Definition 27.** *Let $\mathcal{G}$ be a WCFG, and let $\Delta(\mathcal{G})$ be the set of all derivations of $\mathcal{G}$ rooted at $S$. The* ***allsum*** *$Z(\mathcal{G})$ of $\mathcal{G}$ is the total weight of all derivations of $\mathcal{G}$ rooted at $S$,*

$$Z(\mathcal{G}) \overset{\text{def}}{=} \bigoplus_{d \in \Delta(\mathcal{G})} \mathbf{w}(d). \tag{16}$$

The allsum algorithm (Algorithm 21) computes a table $A$ indexed by $V$, where $A[X]$ represents the total weight of all derivations rooted at $X$. In particular, the value $Z(\mathcal{G}) = A[S]$, which we will sometimes denote by $\boldsymbol{z}$, corresponds to the allsum of $\mathcal{G}$. However, we are interested in derivations of $\mathcal{G}$ of some fixed length. To compute this, we employ the binning semiring (Snæbjarnarson et al., 2025). We therefore lift $\mathcal{G}'$ into the $n_{\max}^{\text{th}}$-order binning semiring (Algorithm 20).[7] When the

---

[7]Notice that this algorithm both converts the original CFG $\mathcal{G}$ into a PCFG $\mathcal{G}'$, and then lifts $\mathcal{G}'$ into the a WCFG $\mathcal{G}_D$ over the $n_{\max}^{\text{th}}$-order binning semiring.

---

**Algorithm 14** Algorithm for sampling a random context-free grammar in Chomsky normal form. Here, $\mu_\Sigma \geq 1$ is the mean number of terminals, $\mu_V \geq 1$ is the mean number of variables, $\mu_{R_b} \geq 1$ is the mean number of binary rules, $\mu_{R_l} \geq 1$ is the mean number of lexical rules, $\mu_c \geq 0$ is the mean number of self-embedding chains, and $\mu_k \geq 2$ is the mean chain length.

---

1. **def** SAMPLECFG($\mu_V, \mu_\Sigma, \mu_{R_b}, \mu_{R_l}, \mu_c, \mu_k$):
2.    sample $|V| \sim \text{NB}(\mu_V - 1, 0.5) + 1$
3.    sample $|\Sigma| \sim \text{NB}(\mu_\Sigma - 1, 0.5) + 1$
4.    sample $|R_b| \sim \text{NB}(\mu_{R_b} - 1, 0.5) + 1$
5.    sample $|R_l| \sim \text{NB}(\mu_{R_l} - 1, 0.5) + 1$
6.    let $V = \{S, X_2, \ldots, X_{|V|}\}$
7.    let $\Sigma = \{a_1, \ldots, a_{|\Sigma|}\}$
8.    let $R_b = \emptyset$
9.    let $R_l = \emptyset$
10.   let $R'_b = \{X \to YZ \mid X, Y, Z \in V\}$
11.   let $R'_l = \{X \to a \mid X \in V, a \in \Sigma\}$
12.   **for** $|R_b|$ times :
13.      sample $X \to YZ$ from $R'_b$ without replacement
14.      add $X \to YZ$ to $R_b$
15.   **for** $|R_l|$ times :
16.      sample $X \to a$ from $R'_l$ without replacement
17.      add $X \to a$ to $R_l$
18.   let $R = R_b \cup R_l$
19.   let $\mathcal{G} = (V, \Sigma, R, S)$ be a CFG
20.   $\mathcal{G} = (V, \Sigma, R, S) \leftarrow \text{TRIMCFG}(\mathcal{G})$
21.   **if** $|V| > 1$ :
22.      sample $c \sim \text{NB}(\mu_c, 0.5)$
23.      **for** $c$ times :
24.        $\mathcal{G} \leftarrow \text{ADDSELFEMBEDDINGCHAIN}(\mathcal{G}, \mu_k)$
25.   **return** $\mathcal{G}$

---

**Algorithm 15** Algorithm for adding a self-embedding chain of rules to a CFG $\mathcal{G}$ in Chomsky normal form. Here, $\mu_k \geq 2$ is the mean derivation length.

---

1. **def** ADDSELFEMBEDDINGCHAIN($\mathcal{G} = (V, \Sigma, R, S), \mu_k$):
2.   sample $k \sim \text{NB}(\mu_k - 2, 0.5) + 2$          ▷*Long-tail distribution with a mean of $\mu_k$*
3.   sample $k_L \sim \mathcal{U}(\{1, \ldots, k-1\})$      ▷*Number of rules that generate symbols to the left.*
4.   let $\phi_1, \ldots, \phi_k$ be a uniformly sampled random permutation of $k_L$ $\top$'s and $(k - k_L)$ $\bot$'s
5.   $V' \leftarrow V \setminus \{S\}$
6.   sample $X_0 \sim \mathcal{U}(V')$
7.   $X \leftarrow X_0$
8.   **for** $i = 1, \ldots, k$ :
9.      **if** $i < k$ :
10.        sample $Y \sim \mathcal{U}(V')$
11.      **else**
12.        $Y \leftarrow X_0$
13.      sample $Z \sim \mathcal{U}(V')$
14.      **if** $\phi_i$ :
15.        add $X \to ZY$ to $R$             ▷*Generate something to the left.*
16.      **else**
17.        add $X \to YZ$ to $R$            ▷*Generate something to the right.*
18.      $X \leftarrow Y$
19.   **return** $\mathcal{G}$

---

allsum algorithm is applied to a WCFG $\mathcal{G}_D$ over the $n_{\max}$th-order binning semiring, the vector $A[X] = (\boldsymbol{v}_0, \boldsymbol{v}_1, \cdots, \boldsymbol{v}_{n_{\max}})$ stores at each position $i \in [0, n_{\max}]$ the total weight of all derivations of $\mathcal{G}'$ of the form $X \overset{*}{\Rightarrow} \boldsymbol{w}$, where $|\boldsymbol{w}| = i$.

Using these quantities, we can renormalize the weights of $\mathcal{G}'$'s derivations for each length $n \in [n_{\min}, n_{\max}]$ such that they form a valid probability distribution. This will give us the

---

**Algorithm 16** Convert a partial DFA $\mathcal{A}$ to a WDFA $\mathcal{A}_D$ over the $n_{\max}$th-order binning semiring with respect to the log semiring.

1. **def** LIFTWEIGHTSDFA($\mathcal{A} = (Q, \Sigma, \delta, q_0, F), n_{\max}$)):
2.    let $\mathcal{A}_D = (Q, \Sigma, \delta', q_0, \rho)$ be a new WDFA over the $n_{\max}$th-order binning semiring with respect to the log semiring
3.    **for** $q \in Q$ :
4.      $k \leftarrow 0$
5.      **for** $q \xrightarrow{a} r \in \delta$ :
6.        $k \leftarrow k + 1$
7.      **if** $q \in F$ :
8.        $k \leftarrow k + 1$
9.      $p \leftarrow -\log k$             ▷*Set the probability to $\frac{1}{k}$ (in log space)*
10.     **for** $q \xrightarrow{a} r \in \delta$ :
11.       add $q \xrightarrow{a/(-\infty, p, -\infty, \ldots, -\infty)} r$ to $\delta'$
12.     **if** $q \in F$ :
13.       $\rho(q) \leftarrow (p, -\infty, \ldots, -\infty)$
14.     **else**
15.       $\rho(q) \leftarrow (-\infty, \ldots, -\infty)$
16.    **return** $\mathcal{A}_D$

---

**Algorithm 17** Weight pushing on $\mathcal{A}_D$, where $\mathcal{A}_D$ is a WDFA over the $D$th-order binning semiring with respect to the log semiring. Given $\mathcal{A}_D$, produce a WDFA $\mathcal{A}'_D$ over the $D$th-order binning semiring with respect to the real semiring that is suitable for length-constrained sampling (Algorithm 19). Also return the allsum weight $z$, which can be used to compute the set of valid lengths (Algorithm 18).

1. **def** PUSHWEIGHTS($\mathcal{A}_D = (Q, \Sigma, \delta, q_0, \rho)$):
2.    $\beta \leftarrow$ BACKWARD($\mathcal{A}_D$)
3.    let $\mathcal{A}'_D = (Q, \Sigma, \delta', q_0, \rho')$ be a new WDFA over the $D$th-order binning semiring with respect to the real semiring
4.    **for** $q \in Q$ :
5.      let $T$ be an empty mapping from $\Sigma$ to $(\mathbb{R} \cup \{-\infty\})^{D+1}$
6.      **for** $q \xrightarrow{a/v} r \in \delta$ :
7.        $T[a] \leftarrow v \otimes \beta[r]$
8.      $T \leftarrow \operatorname*{softmax}_a T[a, :]$   ▷*Convert log probabilities to normalized probabilities. This may safely return NaN for columns with all $-\infty$.*
9.      **for** $q \xrightarrow{a/v} r \in \delta$ :
10.       add $q \xrightarrow{a/T[a]} r$ to $\delta'$
11.    $z \leftarrow \beta[q_0]$
12.    **return** $(\mathcal{A}'_D, z)$

---

conditional distribution $p_{\mathcal{G}'}(w \mid |w| = n)$. To achieve this, we take advantage of the decomposable structure of CFG derivations in CNF. Any partial derivation $d$ in CNF is of the type (1) $d = X \underbrace{\xrightarrow{r} YZ \xRightarrow{*} w_1 Z}_{d_1} \underbrace{\xRightarrow{*} w_1 w_2}_{d_2}$, where $r = X \to YZ$, $d_1$ and $d_2$ are derivations rooted at $Y$ and $Z$, respectively, and $|w_1| = i$ and $|w_2| = n - i$, for $i \in [1, n-1]$, or (2) $d = X \xRightarrow{r} a$, where $r = X \to a$. When $X = S$, we also have the type (3) $d = S \xRightarrow{r} \varepsilon$, where $r = S \to \varepsilon$. For derivations $d$ of type (2) and (3), we simply require $r$'s weight to be non-zero for $d$ to be valid. For derivations of type (1), we also require the weights of derivations $d_1$ and $d_2$ to be non-zero for any split point $i$. The total weights of such derivations was exactly the output of Algorithm 21. Therefore, we first compute the set of all valid derivations for each $X$ and length $n$ (Algorithm 22), and then renormalize their weights (Algorithm 23).

**Algorithm 18** Given an allsum weight $z \in (\mathbb{R} \cup \{-\infty\})^{D+1}$, a minimum length $n_{\min}$, and a maximum length $n_{\max}$, where $D \geq n_{\max}$, compute the set of valid lengths $N$. The allsum weight $z$ can be the second output of Algorithm 17 or Algorithm 21.

1. **def** COMPUTEVALIDLENGTHS($z, n_{\min}, n_{\max}$):
2.   **return** $\{n \in \{n_{\min}, \ldots, n_{\max}\} \mid z_n > -\infty\}$

---

**Algorithm 19** Sample a string of length $n$ from a WDFA $\mathcal{A}'_D$ over the $D^{\text{th}}$-order binning semiring with respect to the real semiring, where $D \geq n$. The DFA $\mathcal{A}'_D$ must be the first output of Algorithm 17.

1. **def** SAMPLE($\mathcal{A}'_D = (Q, \Sigma, \delta, q_0, \rho), n$):
2.   $q \leftarrow q_0$
3.   $w \leftarrow \varepsilon$
4.   **for** $i = n, \ldots, 1$ :
5.     sample $(a, r) \sim p$, where $p((a, r)) = v_i$ for $q \xrightarrow{a/v} r \in \delta$
6.     $q \leftarrow r$
7.     $w \leftarrow wa$
8.   **return** $w$

---

To sample a string of length $n$ (Algorithm 24), we start with a derivation rooted at $S$, and we sample recursively partial derivations according to the weights computed previously. Whenever we sample a partial derivation of type (1), we essentially sample the roots $Y$ and $Z$ of the partial derivations $d_1$ and $d_2$, and a split point $i \in [1, n - i]$.

---

**Algorithm 20** Convert a CFG $\mathcal{G}$ in CNF to a WCFG $\mathcal{G}_D$ over the $n_{\max}^{\text{th}}$-order binning semiring with respect to the log semiring. This implicitly creates an intermediate PCFG $\mathcal{G}'$ with uniform probabilities over the rules for each variable on the left-hand-side. Time complexity: $O(|R| \times n_{\max})$.

1. **def** LIFTWEIGHTSCFG($\mathcal{G} = (V, \Sigma, R, S)$):
2.   let $\mathcal{G}_D = (V, \Sigma, R, S))$ be a new WCFG over the $n_{\max}^{\text{th}}$-order binning semiring with respect to the log semiring
3.   **for** $X \in V$ :
4.     $k \leftarrow 0$
5.     **for** $X \rightarrow \alpha \in R$ :
6.       $k \leftarrow k + 1$
7.     $p \leftarrow -\log k$         ▷*Set the probability to $\frac{1}{k}$ (in log space)*
8.     **for** $X \rightarrow \alpha$ :
9.       **if** $\alpha = a$ :
10.         set $R(X \rightarrow \alpha) = (-\infty, p, -\infty, \ldots, -\infty)$
11.       **else**
12.         set $R(X \rightarrow \alpha) = (p, -\infty, \ldots, -\infty)$
13.   **return** $\mathcal{G}_D$

---

The steps detailed above can be summarized as follows. Given a *trim* CFG $\mathcal{G}$, we run the following pre-processing steps once, before sampling. The CFG $\mathcal{G}$ being trim is essential for the correctness of the procedure, otherwise the PCFG $\mathcal{G}'$ might lose probability mass on invalid derivations.

1. Convert $\mathcal{G}$ into a PCFG $\mathcal{G}'$ by assigning uniform probabilities to the rules for each variable on the left-hand-side, then convert $\mathcal{G}'$ into a WCFG $\mathcal{G}_D$ over the $n_{\max}^{\text{th}}$-order binning semiring with respect to the log semiring (Algorithm 20). Time complexity: $O(|R| \times n_{\max})$.

2. Compute the per-variable derivation weights and allsum $Z(\mathcal{G}_D)$ by running the allsum algorithm on $\mathcal{G}_D$ (Algorithm 21). Time complexity:

3. Compute the set of all valid partial derivations for each variable $X$ and length $n \in [n_{\min}, n_{\max}]$ (Algorithm 22). Time complexity: $O(|R| \times D^2)$.

4. Renormalize the weights of all valid partial derivations for each variable $X$ and length $n \in [n_{\min}, n_{\max}]$ (Algorithm 23). Time complexity: $O(|R| \times D^2)$.

**Algorithm 21** Fixed-point iteration algorithm for computing the allsum of a WCFG $\mathcal{G}$ over the semiring $\mathcal{W}$. Time complexity: $O(|R| \times n_{\text{iter}})$, where $n_{\text{iter}}$ is the number of iterations required for convergence.

1. **def** COMPUTEALLSUM($\mathcal{G} = (V, \Sigma, R, S), \mathcal{W} = (\mathbb{K}, \oplus, \otimes, \mathbf{0}, \mathbf{1})$):
2.   let $A^{(0)}$ be a mapping from $V \cup \Sigma$ to $\mathbb{K}$
3.   **for** $X \in V$:
4.     $A^{(0)}[X] \leftarrow \mathbf{0}$
5.   **for** $a \in \Sigma$:
6.     $A^{(0)}[a] \leftarrow \mathbf{1}$
7.   **repeat**
8.     $n \leftarrow n + 1$
9.     let $A^{(n)}$ be a mapping from $V \cup \Sigma$ to $\mathbb{K}$
10.     **for** $X \in V$:
11.       $A^{(n)}[X] \leftarrow \mathbf{0}$
12.     **for** $a \in \Sigma$:
13.       $A^{(n)}[a] \leftarrow \mathbf{1}$
14.     **for** $X \to \alpha \in R$:
15.       $A^{(n)}[X] \leftarrow A^{(n)}[X] \oplus R(X \to \alpha) \otimes \bigotimes_{\beta \in \alpha} A^{(n-1)}[\beta]$
16.   **until** $A^{(n)} \approx A^{(n-1)}$
17.   $\boldsymbol{z} \leftarrow A[S]$
18.   **return** $A^{(n)}, \boldsymbol{z}$

5. Compute the subset of valid lengths $N \subseteq [n_{\min}, n_{\max}]$ (Algorithm 18). Time complexity: $O(n_{\max} - n_{\min})$.

Then, we run the following steps once per string.

1. Sample a length $n$ uniformly from the set of valid lengths $N$. Time complexity: $O(1)$.

2. Sample a string of length $n$ (Algorithm 24). Time complexity: $O(n)$.

**Algorithm 22** Given a WCFG $\mathcal{G}_D$ over the $D^{\text{th}}$-order binning semiring with respect to the log semiring, a binning size $D$ and an allsum table $A$, compute for each partial derivation rooted at $X$ and length $i \in [0, D]$ a set of valid partial derivations to sample recursively and their weights. Notice that these weights need to be normalized to form a probability distribution. Time complexity: $O(|R| \times D^2)$.

1. **def** COMPUTEVALIDDERIVATIONS($\mathcal{G}_D = (V, \Sigma, R, S), D, A$):
2.   let $T^{(0)}$ be a mapping from $V$ to sets of tuples $\mathbb{K} \times \{\varepsilon\}$
3.   **if** $S \xrightarrow{v} \varepsilon \in R$:
4.     add $(\boldsymbol{v}_0, \varepsilon)$ to $T^{(0)}[S]$
5.   let $T^{(1)}$ be a mapping from $V$ to sets of tuples $\mathbb{K} \times \Sigma$
6.   **for** $X \xrightarrow{v} a \in R$:
7.     add $(\boldsymbol{v}_1, a)$ to $T^{(1)}[X]$
8.   let $T^{(i)}$ for $i = 2 \ldots D$ be a mapping from $V$ to sets of tuples $\mathbb{K} \times (V \times \mathbb{Z}_{\geq 0})^2$
9.   **for** $X \xrightarrow{v} YZ \in R$:
10.     **for** $i = 2 \ldots D$:
11.       **for** $j = 1 \ldots i - 1$:
12.         $w \leftarrow \boldsymbol{v}_0 + A[Y]_j + A[Z]_{i-j}$
13.         add $(w, ((Y, j), (Z, i - j)))$ to $T^{(i)}[X]$
14.   **return** $T^{(0)}, \ldots, T^{(D)}$

---

**Algorithm 23** Normalize the weights of the valid derivation subtrees. The input is one of the mappings produced by Algorithm 22. Time complexity: $O(|R| \times D)$.

---

1. **def** NORMALIZEDERIVATIONWEIGHTS($T^{(i)}$):
2.   **for** $X \in T^{(i)}$ :
3.     $T^{(i)}[X] \leftarrow \mathrm{softmax}_\alpha T^{(i)}[X][:, \alpha]$
4.   **return** $T^{(i)}$

---

---

**Algorithm 24** Sample a string of length $n$ from a WCFG $\mathcal{G}$ over the $D^{\text{th}}$-order binning semiring with respect to the real semiring, where $D \geq n$. Time complexity: $O(n)$

---

1. **def** SAMPLE($\mathcal{G} = (V, \Sigma, R, S), n$):
2.   $w \leftarrow$ SAMPLEDERIVATION($S, n$)
3.   **return** $w$

---

## E   NEURAL NETWORK ARCHITECTURES

We use the same neural network architectures as Butoi et al. (2025). We omit formal definitions of the architectures, but we restate some important details that influence the size of the models and experiments.

### E.1   RNN

We apply dropout to the input embeddings, the hidden states between layers, and the hidden states output from the last layer (Zaremba et al., 2015). The initial hidden state of each layer is learned.

### E.2   LSTM

As with the simple RNN, we apply dropout following Zaremba et al. (2015). The initial hidden state of each layer is learned.

### E.3   TRANSFORMER

Following Vaswani et al. (2017), we map input symbols to vectors of size $d$ with a scaled embedding layer and add sinusoidal positional encodings. We use 8 attention heads in each layer, and we set the number of hidden units in each feedforward layer to $4d$. We use pre-norm instead of post-norm and apply layer norm to the output of the last layer. We use the same dropout rate throughout the transformer. We apply it in the same places as Vaswani et al. (2017), and, as implemented by PyTorch, we also apply it to the hidden units of feedforward sublayers and to the attention probabilities of scaled dot-product attention operations. We always use BOS as the first input symbol to the transformer, which has been shown to improve performance on formal languages (Ebrahimi et al., 2020).

## F   EXPERIMENTAL DETAILS

Here, we provide additional details about the models and training procedures used in §5. When applicable, we apply dropout with a rate of 0.1. For the transformer, when adjusting the hidden dimension to accommodate the parameter budget, we round it to the nearest multiple of 8, since PyTorch requires the dimension to be divisible by the number of attention heads. For layer norm, we initialize weights to 1 and biases to 0. All remaining parameters are initialized by sampling from a uniform distribution over $[-0.1, 0.1]$. During each epoch, we shuffle the training set and form minibatches by grouping together strings of similar lengths. Each batch is constrained to contain at most $B$ symbols in total, including padding, BOS and EOS symbols. Models are trained with the loss defined in Eq. (4) using Adam (Kingma & Ba, 2015). We apply gradient clipping with an $L^2$-norm threshold of 5. At the end of every epoch, we save a checkpoint, evaluate the model on the validation set, and update both the learning-rate schedule and early-stopping schedules. The learning rate is

---

**Algorithm 25** Sample a derivation tree rooted at $X$ that generates a string of length $n$. Time complexity: $O(n)$

---

1. **def** $\text{SAMPLEDERIVATION}(X, n)$:
2.    **if** $n = 0$ or $n = 1$ :
3.      sample $a \sim p$ where $p(a) = w$ for $(a, w)$ in $T^{(n)}[X]$
4.      **return** $a$
5.    **else**
6.      sample $(Y, i), (Z, j) \sim p$ where $p((Y, i), (Z, j)) = w$ for $((Y, i), (Z, j), w)$ in $T^{(n)}[X]$
7.      $\boldsymbol{w}_1 \leftarrow \text{SAMPLEDERIVATION}(Y, i)$
8.      $\boldsymbol{w}_2 \leftarrow \text{SAMPLEDERIVATION}(Z, j)$
9.      **return** $\boldsymbol{w}_1 \boldsymbol{w}_2$

---

halved if validation cross-entropy does not improve for 5 consecutive checkpoints, and training is stopped early if there is no improvement for 10 consecutive checkpoints. For evaluation, we report results from the checkpoint achieving the lowest validation cross-entropy. We train for a maximum of 1k epochs if there is no early stopping. Every time we train a model, we randomly sample a number of hyperparameters. We randomly sample the batch size from a uniform distribution over $[128, 4096]$, and the initial learning rate from a log-uniform distribution over $[0.0001, 0.01]$.

## G  PERFORMANCE VS. MACHINE SIZE

We show accuracy vs. the size of the automaton/grammar generating the language for the models shown under "Expressivity" in Table 1. For each language class, we plot the accuracy of the best-performing model for the 20 languages. The curves are smoothed for the sake of readability. In most cases, we observe a decrease in performance as the size of the machine increases. We report the accuracy as a function of alphabet size and number of states for PODFA (Figure 2) and regular languages (Figure 4), as a function of alphabet size, number of states, and dot-depth for star-free languages (Figure 3), and as a function of alphabet size, number of variables, and number of rules for context-free languages (Figure 5).

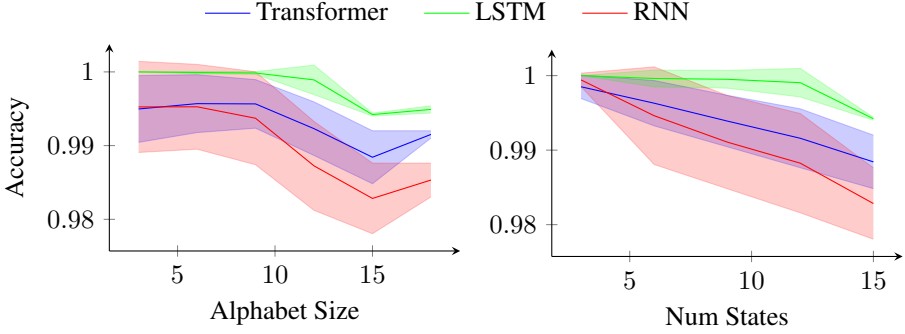

Figure 2: Average accuracy vs. automaton size across the 20 PODFA languages shown under "Expressivity" in Table 1.

## H  PERFORMANCE VS. LENGTH OF THE INPUT SEQUENCE

For each language class, we show the average cross entropy vs. the length of the input sequence for the best-performing models trained for the expressivity experiments (Figure 6). The curves are smoothed for the sake of readability. For star-free languages, the cross entropy of the transformer and LSTM increases sharply outside the training range. Similarly, the cross entropy of all architectures increases slightly on strings with lengths outside the training range on regular languages.

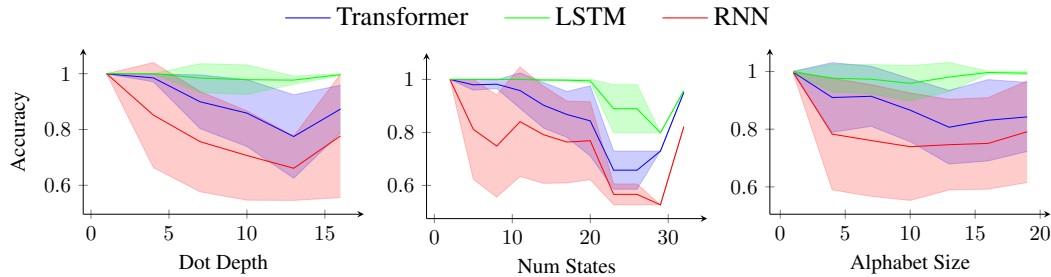

Figure 3: Average accuracy vs. automaton size across the 20 star-free languages shown under "Expressivity" in Table 1.

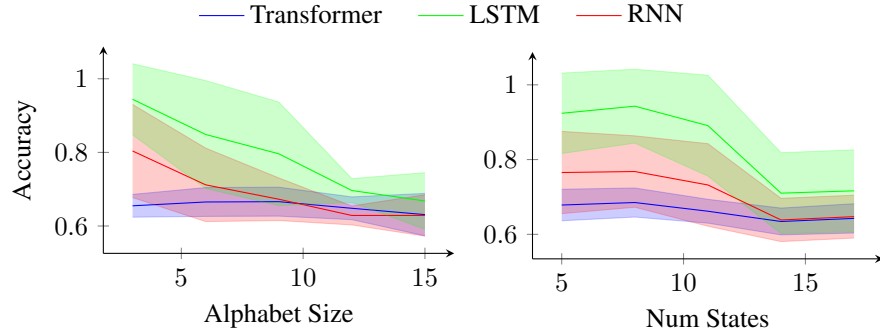

Figure 4: Average accuracy vs. automaton size across the 20 regular languages shown under "Expressivity" in Table 1.

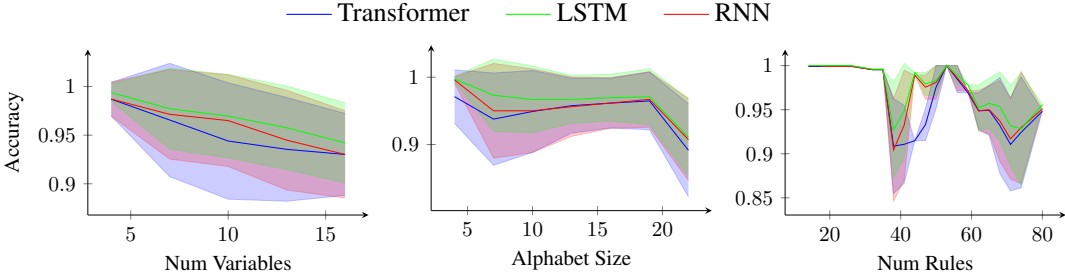

Figure 5: Average accuracy vs. automaton size across the 20 context-free languages shown under "Expressivity" in Table 1.

## I  PERFORMANCE VS. DATASET SIZE

For PODFA and regular languages, we vary the dataset size to check whether training on more data leads to improved performance compared to that reported in §6. In this set of experiments, we use a long validation set with strings in the range $[0, 80]$. We train models with different numbers of parameters (128k, 256k, 512k, and 1024k) for 10 languages from each language class. For each language, architecture, and parameter budget, we train 10 models with different hyperparameters (e.g., batch size, initial learning rate). We show in Table 3 the average accuracy over the best-performing trials per language, and in Table 4 the percentage of languages learned perfectly, i.e., languages for which there exists at least one trial with perfect accuracy.

We observe that adding more training data improves performance in several cases. For regular languages, increasing the dataset from 10k to 100k examples substantially boosts the LSTM's accuracy, from 82% to 98%, and increases the proportion of languages it learns perfectly from 20%

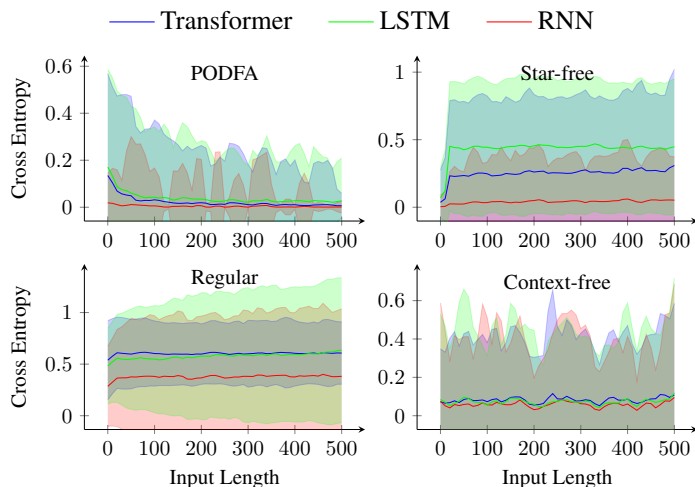

Figure 6: Average cross entropy vs. input length across the 20 languages from each language class.

Table 3: Accuracy achieved by transformers, RNNs, and LSTMs on 10 regular and partially-ordered DFA (PODFA) languages as a function of dataset size. For each language, we train 10 models with different parameter settings. The models are trained on strings with lengths in the range $[0, 40]$ and tested on strings with lengths in the range $[0, 500]$. We test expressivity by using validation sets containing strings with lengths in the range $[0, 80]$. We report the average accuracy over the best-performing trials per language.

| | PODFA | | | Regular | | |
|---|---|---|---|---|---|---|
| Dataset Size | Tf | RNN | LSTM | Tf | RNN | LSTM |
| 10k | $0.99_{\pm 0.00}$ | $1.00_{\pm 0.01}$ | $1.00_{\pm 0.00}$ | $0.67_{\pm 0.05}$ | $0.69_{\pm 0.08}$ | $0.82_{\pm 0.15}$ |
| 20k | $0.99_{\pm 0.00}$ | $0.99_{\pm 0.00}$ | $1.00_{\pm 0.00}$ | $0.68_{\pm 0.04}$ | $0.71_{\pm 0.08}$ | $0.84_{\pm 0.13}$ |
| 50k | $0.99_{\pm 0.00}$ | $0.99_{\pm 0.01}$ | $1.00_{\pm 0.00}$ | $0.69_{\pm 0.04}$ | $0.72_{\pm 0.09}$ | $0.95_{\pm 0.10}$ |
| 100k | $0.99_{\pm 0.00}$ | $0.99_{\pm 0.01}$ | $1.00_{\pm 0.00}$ | $0.69_{\pm 0.05}$ | $0.72_{\pm 0.08}$ | $0.98_{\pm 0.05}$ |

Table 4: Percentage of regular and partially-ordered DFA (PODFA) languages learned perfectly by transformers, RNNs, and LSTMs as a function of dataset size. For each architecture and language class, we train 10 models on each of the 10 languages with different hyperparameter settings. For each language class and architecture, we report the percentage of languages on which at least one of the models achieves perfect accuracy. The models are trained on strings with lengths in the range $[0, 40]$ and tested on strings with lengths in the range $[0, 500]$. We test expressivity by using validation sets containing strings with lengths in the range $[0, 80]$.

| | PODFA | | | Regular | | |
|---|---|---|---|---|---|---|
| Language | Tf | RNN | LSTM | Tf | RNN | LSTM |
| 10k | 0% | 20% | 70% | 0% | 0% | 20% |
| 20k | 0% | 30% | 100% | 0% | 0% | 40% |
| 50k | 0% | 30% | 100% | 0% | 0% | 50% |
| 100k | 10% | 30% | 100% | 0% | 0% | 90% |

to 90%. For PODFA languages, the LSTM achieves near-perfect accuracy regardless of model size, but increasing the dataset size increases the percentage of perfectly learned languages from 70% to 100%. For the other architectures, the gains are more modest: on regular languages, the transformer's accuracy rises from 67% to 69%, and RNN's accuracy increases from 69% to 72%. On PODFA languages, the proportion of languages learned perfectly increases by 10% for both the transformer and the RNN.

Table 5: Accuracy achieved by transformers, RNNs, and LSTMs on 10 regular and partially-ordered DFA (PODFA) languages as a function of number of parameters. For each language class, architecture, and parameter budget, we train 10 models with different parameter settings. The models are trained on strings with lengths in the range $[0, 40]$ and tested on strings with lengths in the range $[0, 500]$. We test expressivity by using validation sets containing strings with lengths in the range $[0, 80]$. For each language class, architecture, and parameter budget, we report the average accuracy over the best-performing trials.

| Language | PODFA | | | Regular | | |
|---|---|---|---|---|---|---|
| | Tf | RNN | LSTM | Tf | RNN | LSTM |
| 128k | $0.99_{\pm 0.01}$ | $0.99_{\pm 0.01}$ | $1.00_{\pm 0.00}$ | $0.69_{\pm 0.05}$ | $0.72_{\pm 0.08}$ | $0.97_{\pm 0.07}$ |
| 256k | $0.99_{\pm 0.00}$ | $0.99_{\pm 0.01}$ | $1.00_{\pm 0.00}$ | $0.69_{\pm 0.05}$ | $0.71_{\pm 0.09}$ | $0.98_{\pm 0.07}$ |
| 512k | $0.99_{\pm 0.00}$ | $0.99_{\pm 0.01}$ | $1.00_{\pm 0.00}$ | $0.69_{\pm 0.05}$ | $0.70_{\pm 0.10}$ | $0.98_{\pm 0.07}$ |
| 1024k | $0.99_{\pm 0.00}$ | $0.99_{\pm 0.01}$ | $1.00_{\pm 0.00}$ | $0.69_{\pm 0.05}$ | $0.68_{\pm 0.09}$ | $0.98_{\pm 0.05}$ |

## J  PERFORMANCE VS. MODEL SIZE

For PODFA and regular languages, we asses the performance of the three architectures as a function of model size. In this set of experiments, we train the models on datasets containing 100k examples. We test expressivity by using a long validation set with strings in the range $[0, 80]$. We train models with different numbers of parameters (128k, 256k, 512k, and 1024k) for 10 languages from each language class. For each language, architecture, and parameter budget, we train 10 models with different hyperparameter settings. We show for each parameter budget the average accuracy over the best-performing trials per language, and the percentage of languages learned perfectly, i.e., languages for which there exists at least one trial with perfect accuracy.

In general, the performance stays approximately the same when varying the model size (Table 5, Table 6). We only notice a slight increase in accuracy for the LSTM on regular languages, by approximately 1%, and a 4% decrease in accuracy for the RNN on regular languages.

Table 6: Percentage of regular and partially-ordered DFA (PODFA) languages learned perfectly by transformers, RNNs, and LSTMs as a function of number of parameters. For each architecture, language class, and parameter budget, we train 10 models each on 10 languages, with different hyperparameter settings. We report the percentage of languages on which at least one of the models achieves perfect accuracy. The models are trained on strings with lengths in the range $[0, 40]$ and tested on strings with lengths in the range $[0, 500]$. We test expressivity by using validation sets containing strings with lengths in the range $[0, 80]$.

| Language | PODFA | | | Regular | | |
|---|---|---|---|---|---|---|
| | Tf | RNN | LSTM | Tf | RNN | LSTM |
| 128k | 0% | 20% | 90% | 0% | 0% | 70% |
| 256k | 10% | 30% | 100% | 0% | 0% | 70% |
| 512k | 0% | 30% | 100% | 0% | 0% | 90% |
| 1024k | 0% | 20% | 100% | 0% | 0% | 70% |

## K  BASELINES

We train a set of baseline models, including an n-gram model, an SVM, and logistic regression, to measure the inherent difficulty of languages within each language class and to provide reference points for our model evaluations.

Table 7: Average accuracy achieved by an n-gram model, an SVM, and logistic regression on the same 20 languages from each language class as those shown in Table 1.

| Language class | Accuracy | | |
| --- | --- | --- | --- |
| | n-gram | SVM | Logistic regression |
| PODFA | $0.81_{\pm 0.02}$ | $0.81_{\pm 0.02}$ | $0.81_{\pm 0.02}$ |
| SF | $0.60_{\pm 0.13}$ | $0.60_{\pm 0.16}$ | $0.60_{\pm 0.16}$ |
| R | $0.74_{\pm 0.03}$ | $0.65_{\pm 0.06}$ | $0.64_{\pm 0.06}$ |
| CF | $0.90_{\pm 0.06}$ | $0.90_{\pm 0.06}$ | $0.90_{\pm 0.06}$ |

The n-gram model trains a positive and a negative model on the training data, using n-gram counts. For each class, it first counts the number of occurrences of each n-gram, then computes the smoothed probabilities

$$P(\boldsymbol{w}_{i-n+1}\cdots \boldsymbol{w}_i \mid \text{class}) = \frac{\text{count}(\boldsymbol{w}_{i-n+1}\cdots \boldsymbol{w}_i, \text{class}) + \alpha}{N + \alpha|V|}, \tag{17}$$

where $N$ is the total number of n-grams in the class, and $|V|$ is the size of the vocabulary on n-grams in the class. Then, for a test string $\boldsymbol{w} = \boldsymbol{w}_1 \cdots \boldsymbol{w}_n$, it computes the log-likelihood under each model,

$$\log P(\boldsymbol{w} \mid \text{class}) = \sum_{i=n}^{n} \log P(\boldsymbol{w}_{i-n+1}\cdots \boldsymbol{w}_i \mid \text{class}), \tag{18}$$

and predicts a label using the likelihood ratio

$$\begin{cases} 1, & \log P(\boldsymbol{w} \mid \text{positive}) - \log P(\boldsymbol{w} \mid \text{negative}) > 0, \\ 0, & \text{otherwise.} \end{cases} \tag{19}$$

We train the SVM model and logistic regression with n-gram features. Each string $\boldsymbol{w}$ is converted into a vector $\mathbf{c} = (\mathbf{c}_1, \mathbf{c}_2, \cdots, \mathbf{c}_k)$, where $c_i$ is the count of the $i^{\text{th}}$ n-gram in the vocabulary. The SVM model learns a hyperplane $\mathbf{v}^\mathsf{T}\mathbf{c} + b = 0$ such that $\mathbf{v}$ and $b$ maximize the margin between the positive and negative classes. Then, on a test string, it predicts

$$\begin{cases} 1, & \mathbf{v}^\mathsf{T}\mathbf{c} + b \geq 0, \\ 0, & \mathbf{v}^\mathsf{T}\mathbf{c} + b < 0. \end{cases} \tag{20}$$

The logistic regression model learns weights $\mathbf{v} = (\mathbf{v}_1, \mathbf{v}_2, \cdots, \mathbf{v}_k)$ and the bias $b$. Then, on a test string, the model computes $z = \mathbf{v}_1\mathbf{c}_1 + \mathbf{v}_2\mathbf{c}_2 + \ldots + \mathbf{v}_k\mathbf{c}_k + b$ and outputs a prediction using $P(1 \mid \boldsymbol{w}) = \sigma(z)$.

For all models, we experimented with multiple values of $n$ for the n-grams and we observed very similar accuracies. Table 7 shows results with $n = 3$ for the n-gram model, and $n = 1, 2$ for the SVM and logistic regression, i.e., the features include unigram and bigram counts.

We observe that the performance of the baseline models does not necessarily correlate with the theoretical complexity of the language classes. With the exception of the regular languages, all models obtain very similar accuracies across all classes. Notably, they achieve their highest average accuracy on the context-free languages, the highest on the Chomsky hierarchy, surpassing even their performance on the PODFA languages, which are the simplest. Likewise, the models perform better on regular languages than on star-free languages, despite the fact that star-free languages form a subset of regular languages.

# L    CROSS-ENTROPY VS. EDIT DISTANCE

For context-free languages, we examine how the similarity between negative and positive examples influences recognition difficulty. Specifically, we show in Figure 7 the models' cross-entropy as

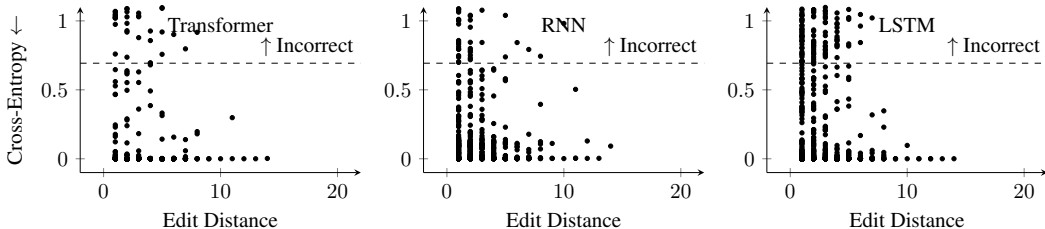

Figure 7: Cross entropy as a function of edit distance for negative strings from three randomly sampled context-free languages.

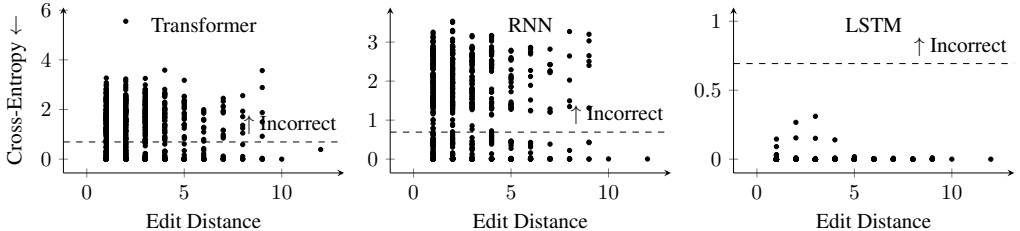

Figure 8: Cross entropy as a function of edit distance for negative strings from three randomly sampled star-free languages.

a function of the edit distance between a perturbed string and the target language (the strings are generated using the perturbation procedure described in §4.2) for three languages.[8] We approximate edit distance using the number of random edits, $K$, which serves as an upper bound on the true edit distance. The cross-entropy is that of the best performing model.

The largest number of errors occurs for negative strings with small edit distances, generally smaller than five, while strings with larger edit distances are almost always assigned low cross-entropy, and thus correctly rejected. This suggests that the models are not capturing the underlying structure of the languages, but rather rely on superficial cues. In contrast, on star-free languages (Figure 8) the models' misclassification rate does not seem to correlate with the edit distance.

---

[8]Because the plot contains a very large number of points, three languages are the maximum we can reasonably display at once.

