# OpenReview forum: "Empirically Testing Expressivity Bounds for Neural Network Architectures"
_ICLR.cc/2026/Conference — Submitted to ICLR 2026_

### Official Review · Reviewer_rgAY · 2025-10-30

**Soundness:** 2
**Presentation:** 3
**Contribution:** 3
**Rating:** 4
**Confidence:** 4

**Summary:**

This paper samples random languages from a set of language families; specifically context-free languages, regular languages, star-free languages with differing dot counts, and regular languages corresponding to partially ordered dfas. A model from one of three classes (LSTM, RNN, Transformer) is trained on each of these languages and average results are reported for both average performance and the percentage of fully solved languages. Additionally, an algorithm is presented to sample strings of a particular length from a context-free grammar. The overall results are that on these distributions, all model classes fail at fully solving most languages, except in the PODFA family, but are nonetheless able to make fairly accurate predictions on all language families, in particular the context-free languages.

**Strengths:**

The technical presentation is generally pretty clear and elementary (See minor weaknesses and questions for some caveats to this), and in general the paper’s theoretical component is very rigorous and sound.

**Weaknesses:**

A significant weakness with this work is the lack of a detailed discussion in the main text of the fact that the sampling procedures are uncalibrated, that is, there is no attempt made to ensure that the distributional choices do not affect the apparent difficulty of each language family. (To take an extreme example, if the regular languages sampled all corresponded to 30 state DFAs, while the CFG sampler always returned the nested parentheses language, then the CFGs would be “simpler” without this saying much about the inherent difficulty of CFGs in general vs regular languages in general). I acknowledge that to some degree, such calibration is impossible, as there are qualitative differences between these spaces so there is no way to ensure that the sampling procedures are identical; however, I do believe that some attempt could be made to at least calibrate via some non-neural method. E.g., the use of some non-neural method such as an n-gram could be used to provide a general estimate of how difficult the distribution sampled from each language family is. I ask this primarily because I find it difficult to interpret the result showing that CFGs are surprisingly easy to learn heuristics for, due to the uneliminated possibility that the sampling procedure you have devised biases towards CFGs that can be approximated with easy-to-learn shallow heuristics. I believe this problem is fundamental to any comparison between language classes (though it does not affect comparisons between model classes).


Minor:

An unfamiliar reader might not realize that star-free languages contain languages that must be represented with stars in regular expressions; this should be clarified in a footnote. I personally had to refresh my memory by looking it up, despite working with automata regularly.

**Questions:**

Definition 13 is the polynomial ring, right? If I am correct, I think you should mention this for clarity’s sake, even if only in a footnote. It might also be clearer to just introduce this as polynomials rather than abstract vectors. I could be wrong and have missed something, in which case, ignore this.

---

> ### Author Response · Authors · 2025-11-14
> **Response to reviewer rgAY**
>
> Thank your very much for your detailed review and comments! We are glad that you found our work rigorous and sound. We address your concerns in the following.
>
> > A significant weakness with this work is the lack of a detailed discussion in the main text of the fact that the sampling procedures are uncalibrated, that is, there is no attempt made to ensure that the distributional choices do not affect the apparent difficulty of each language family. (To take an extreme example, if the regular languages sampled all corresponded to 30 state DFAs, while the CFG sampler always returned the nested parentheses language, then the CFGs would be “simpler” without this saying much about the inherent difficulty of CFGs in general vs regular languages in general). I acknowledge that to some degree, such calibration is impossible, as there are qualitative differences between these spaces so there is no way to ensure that the sampling procedures are identical; however, I do believe that some attempt could be made to at least calibrate via some non-neural method. E.g., the use of some non-neural method such as an n-gram could be used to provide a general estimate of how difficult the distribution sampled from each language family is. I ask this primarily because I find it difficult to interpret the result showing that CFGs are surprisingly easy to learn heuristics for, due to the uneliminated possibility that the sampling procedure you have devised biases towards CFGs that can be approximated with easy-to-learn shallow heuristics. I believe this problem is fundamental to any comparison between language classes (though it does not affect comparisons between model classes).
>
> We are currently working on some new analysis that aims to do exactly this. We will train an n-gram model to check whether the sampled CFLs are indeed simpler. Additionally, we will analyze the model performance as a function of string edit distance.
>
> > An unfamiliar reader might not realize that star-free languages contain languages that must be represented with stars in regular expressions; this should be clarified in a footnote. I personally had to refresh my memory by looking it up, despite working with automata regularly.
>
> Thank you for your suggestion. Star-free languages are those that can be expressed by regular expressions without the star operation. We will add an explanation in the updated version.
>
> > Definition 13 is the polynomial ring, right? If I am correct, I think you should mention this for clarity’s sake, even if only in a footnote. It might also be clearer to just introduce this as polynomials rather than abstract vectors. I could be wrong and have missed something, in which case, ignore this.
>
> There is indeed a connection between Definition 13 and a truncated version of the polynomial ring. We will add an explanation in the new draft.

---

> > ### Comment · Reviewer_rgAY · 2025-11-22
> >
> > Thank you for your response!
> >
> > > We are currently working on some new analysis that aims to do exactly this. We will train an n-gram model to check whether the sampled CFLs are indeed simpler. Additionally, we will analyze the model performance as a function of string edit distance.
> >
> > Glad to hear about this, and will be interested in the results.
> >
> > >  Thank you for your suggestion. Star-free languages are those that can be expressed by regular expressions without the star operation. We will add an explanation in the updated version.
> >
> > Unless I am missing something, this is not true, since they can use complementation, which is not available in traditional regular expressions, as such you can express languages such as “all strings”, which cannot be expressed purely via literals, alternation, and concatenation: in fact, the only languages that can be expressed with these operations are finite.
> >
> > > There is indeed a connection between Definition 13 and a truncated version of the polynomial ring. We will add an explanation in the new draft.
> >
> > Thank you, this will be helpful for understanding.

---

> ### Author Response · Authors · 2025-11-26
> **Response to reviewer rgAY (2)**
>
> We ran some additional experiments and updated our draft to address your comments. We highlighted the changes in blue to make it easier for the reviewers to find these. You might be interested in the following changes:
>
> 1. We trained 3 non-neural baselines to assess the difficulty of the languages sampled from each language class, namely an n-gram model, an SVM and a logistic regression. We mention these in Sections 5 and 6 and give additional details in Appendix K. We observe the following: the performance of the baselines does not necessarily correlate with the complexity of the language class, or the performance of the neural models. Both the neural models and the non-neural baselines obtain very high accuracy on context-free languages, which might suggest that the sampled languages are inherently easier.
>
> 2. We added an additional comment about the star-free languages in lines 210-212. We apologize for the lack of clarity in our previous comment; we meant extended regular expressions.
>
> 3. We added a footnote that explains how the binning semiring is related to the polynomial ring (starting at line 376). Thank your for this suggestion!
>
> We mention that we are still working on some new analysis for the context-free languages, e.g., analysis of the performance as a function of edit distance, and we will try to add these as soon as possible. Given that we made quite a few changes, we wanted to upload a new draft with the results we have so far so that the reviewers have enough time to check the changes before the end of the discussion period.

---

> > ### Author Response · Authors · 2025-11-30
> > **Response to reviewer rgAY (3)**
> >
> > > I ask this primarily because I find it difficult to interpret the result showing that CFGs are surprisingly easy to learn heuristics for, due to the uneliminated possibility that the sampling procedure you have devised biases towards CFGs that can be approximated with easy-to-learn shallow heuristics.
> >
> > We did some additional analysis to address reviewer rgAY's comment (see the discussion from Section 6 and Appendix L in the updated draft). In our original draft, we noticed that all architectures achieved around 95\% average accuracy on context-free languages. However, they mastered only a small subset of these languages perfectly, suggesting that while the models capture useful approximations, they still fail to learn the underlying generative mechanisms. To test this hypothesis, we plotted the cross-entropy of negative strings as a function of edit distance. We found that most misclassifications clustered at small edit distances, suggesting that the models rely on superficial cues rather than capturing the true structure of the target language. For comparison, we generated analogous plots for the star-free languages and observed that, in contrast, misclassifications in these languages are not correlated with edit distance.

---

### Official Review · Reviewer_F4v3 · 2025-10-31

**Soundness:** 2
**Presentation:** 4
**Contribution:** 2
**Rating:** 4
**Confidence:** 3

**Summary:**

This paper provides sampling algorithms for sampling from the set of all languages in various subcategories of regular languages (e.g., star-free languages of varying dot-dept) and for sampling from those resulting languages with varying string length constraints, and tests LSTMs, RNNs, and Transformers on their ability to learn to recognize those languages.

**Strengths:**

I found the presentation of this work accessible, and it gave a nice clear description of background in theoretical expressivity results in neural networks for language recognition. This is a technical subfield, and so, nicely presented exposition is a big plus!

I want to acknowledge that there’s a ton of work that goes into the results provided – sampling from the set of languages, sampling from the length-constrained set of strings for each language – and characterizing the computability or hardness of each. I found the appendix really nice, characterizing a lot of extra formal details; efficiently sampling from a constrained set of lengths under a PCFG isn’t obvious! It seems like a ton of solid work went into getting the formal properties right and I value that highly.

**Weaknesses:**

I think “empirically testing expressivity bounds” is just not what this paper does. It does characterize learnability for a set of hyperparameters and a specific learning algorithm and a specific loss etc. I think the expressivity bounds are what they are – if the proofs are right (and the premises hold for the architectures being used; I realize this is not always the case, e.g., with nonstandard position encodings) – then maybe this is just semantics but we’re really exploring the relationship between expressivity and learnability for these architectures. I’m not arguing that the authors disagree with the proofs in the literature – read on for my point:

This leads to the core weakness here, which is that I’m just not convinced in the empirical results; I see no details as to the size/depth/other hyperparmaeters of each of the networks, nor the learning algorithm, nor the number of samples used for training, nor the number of training steps. Certainly we can never test expressivity by evaluating learnability, but if we want to try to measure how they diverge in practice, we need to work very hard at hyperparameter optimization so that we don’t underestimate the learnability of a language under a given model architecture. In this, relative to the details we find for the definitions and sampling algorithms for the formal languages provided, this paper provides very few details about the effort that went into the learning process. The paper notes that 5 hyperparameter settings were used – which? I looked over the appendix as well and could find no details. And regardless, this seems really low. I really struggle with the claim of “testing expressivity bounds” here – it seems like a ton of work should go into getting the layer count, optimizer, hidden dimensionality right, and we should see learning curves as a function of the number of samples and training steps. This is especially true given the odd optimization behavior of deep networks (e.g., “grokking.”)

This is tough for me because the formal presentation of this work is really strong, and I feel like I learn from that and the discussions around how to sample from families of languages. But the core claims of the work are inherently empirical—can these architectures recognize members of these language families–, and I don’t think these claims are properly supported.

Next, and this is minor – I’m generally in agreement that there’s utility of understanding the connections between classes of formal languages and network expressivity. However, while the authors are right that we shouldn’t “just” focus on specific exemplars of languages of a class like the Dyck-k languages, these exemplars at least are well-motivated through their connections we care about, e.g., natural languages and programming languages. I’d like the authors to do more to motivate why their particular sampling strategies over, e.g., the family of dot-depth 3 languages, lead to languages that we should care about.

**Questions:**

You note that your algorithms for sampling from the set of languages of a given class have every language in the class as their support. Can you characterize though how the likelihood decays, e.g., as a function of complexity? I’m just not sure which star-free languages your sampling methods end up putting, e.g., exponentially small probability on (as a function of number of DFA states for example.)

---

> ### Author Response · Authors · 2025-11-14
> **Response to reviewer F4v3**
>
> Thank your very much for your detailed review and suggestions! We are glad that you found our work nicely presented and solid. We address your comments below.
>
> > I think “empirically testing expressivity bounds” is just not what this paper does. It does characterize learnability for a set of hyperparameters and a specific learning algorithm and a specific loss etc. I think the expressivity bounds are what they are – if the proofs are right (and the premises hold for the architectures being used; I realize this is not always the case, e.g., with nonstandard position encodings) – then maybe this is just semantics but we’re really exploring the relationship between expressivity and learnability for these architectures. I’m not arguing that the authors disagree with the proofs in the literature – read on for my point:
>
> We agree that the title is somewhat misleading. We will change the title and upload a new draft soon.
>
> > This leads to the core weakness here, which is that I’m just not convinced in the empirical results; I see no details as to the size/depth/other hyperparmaeters of each of the networks, nor the learning algorithm, nor the number of samples used for training, nor the number of training steps. Certainly we can never test expressivity by evaluating learnability, but if we want to try to measure how they diverge in practice, we need to work very hard at hyperparameter optimization so that we don’t underestimate the learnability of a language under a given model architecture. In this, relative to the details we find for the definitions and sampling algorithms for the formal languages provided, this paper provides very few details about the effort that went into the learning process. The paper notes that 5 hyperparameter settings were used – which? I looked over the appendix as well and could find no details. And regardless, this seems really low. I really struggle with the claim of “testing expressivity bounds” here – it seems like a ton of work should go into getting the layer count, optimizer, hidden dimensionality right, and we should see learning curves as a function of the number of samples and training steps. This is especially true given the odd optimization behavior of deep networks (e.g., “grokking.”)
>
> Thank you for your comment. We agree that this is not detailed enough, either in the main text or the appendix. Every time we train a model (5 for each language, validation set, and parameter budget in the current draft), we randomly sample the initial learning rate and batch size from a log-uniform distribution over [0.0001,0.01], and a uniform distribution over [128,4096], respectively. We train models for different parameter budgets (128k, 256k, and 512k) and we report the mean across the best-performing trials over all configurations for expressivity, and the mean across all trials for inductive bias. Therefore, we train 30 models for each pair of language and architecture.
>
> We use 5 layers for all models in all experiments. We automatically adjust the hidden vector size so
> that the number of parameters is as close as possible to the predefined parameter budget. Wherever dropout is applicable, we use a dropout rate of 0.1. For each epoch, we randomly shuffle the training set and group strings of similar lengths into the same minibatch. We take a checkpoint every 10k examples, at which point we evaluate the model on the validation set and update the learning rate and early stopping schedules. We multiply the learning rate by 0.5 after 5 checkpoints of no decrease in recognition cross-entropy on the validation set, and we stop early after 10 checkpoints of no decrease. We select the checkpoint with the lowest cross-entropy on the validation set when reporting results. We train for a maximum of 1k epochs if there is no early stopping.
>
> We mention that we reran the experiments with 10 trials per configuration in the meantime and we obtained similar results. We will update the draft with these new results.
>
> (continued in next comment)

---

> ### Author Response · Authors · 2025-11-14
> **Response to reviewer F4v3 (2)**
>
> > Next, and this is minor – I’m generally in agreement that there’s utility of understanding the connections between classes of formal languages and network expressivity. However, while the authors are right that we shouldn’t “just” focus on specific exemplars of languages of a class like the Dyck-k languages, these exemplars at least are well-motivated through their connections we care about, e.g., natural languages and programming languages. I’d like the authors to do more to motivate why their particular sampling strategies over, e.g., the family of dot-depth 3 languages, lead to languages that we should care about.
>
> The PODFA languages are such a subclass. The reason we distinguish them from general star-free languages is because they were recently connected to the expressivity of soft-attention transformers [1]. This comes in contrast to previous work, which showed that hard-attention transformers can express all star-free languages. This distinction gives a natural motivation to compare PODFA languages and more general star-free languages. By sampling and testing within this family, we aim to verify empirically  whether these theoretical expressivity limits manifest in practice, and indeed we found that the results match the theory.
>
> Another example is the context-free languages. While Dyck-k languages capture hierarchical dependencies and are therefore useful toy examples, they represent only a narrow subclass of context-free languages. Real programming languages require the broader expressivity of general CFLs to capture syntactic dependencies beyond nesting.
>
> [1] Jiaoda Li and Ryan Cotterell. Characterizing the expressivity of transformer language models, 2025.
> URL https://arxiv.org/abs/2505.23623.
>
> > You note that your algorithms for sampling from the set of languages of a given class have every language in the class as their support. Can you characterize though how the likelihood decays, e.g., as a function of complexity? I’m just not sure which star-free languages your sampling methods end up putting, e.g., exponentially small probability on (as a function of number of DFA states for example.)
>
> All our algorithms first sample a number of hyperparameters such as automaton size, alphabet size, dot-depth, etc., from a negative binomial distribution. For this reason, the probability decreases as a function of the hyperparameter. We will make sure to include a discussion in the new draft.

---

> > ### Author Response · Authors · 2025-11-26
> > **Response to reviewer F4v3 (3)**
> >
> > We ran some additional experiments and updated our draft to address your comments. We highlighted the changes in blue to make it easier for the reviewers to find these. You might be interested in the following changes:
> >
> > 1. We updated the title. We hope the new title conveys better what our paper does.
> >
> > 2. We added more concrete details about the experimental setup in Section 5 and added further information in Appendix F.
> >
> > 3. We rewrote part of the introduction to include our response regarding why our sampling methods lead to languages of interest. Thank you very much for your question! We believe this strengthens our motivation.

---

### Official Review · Reviewer_rq1a · 2025-11-01

**Soundness:** 3
**Presentation:** 2
**Contribution:** 2
**Rating:** 4
**Confidence:** 4

**Summary:**

This paper presents an empirical study evaluating the ability of Transformers, RNNs, and LSTMs to recognize formal languages.
This work introduces novel methods for randomly sampling languages from different classes in the Chomsky hierarchy (Regular, Star-Free, and Context-Free) and for generating labeled datasets with controlled string lengths. The experiment results reveal that, all architectures perform well on simpler Partially-Ordered DFA (PODFA) languages, and all models achieve high average accuracy on Context-Free languages, though they master very few of them perfectly. The work has its primary contribution in helping mitigate the gap between theoretical representational capacity and empirical learnability, which can use formal language recognition as a mean for probing the representational capabilities of neural networks.

**Strengths:**

1. The paper's primary strength lies in testing on a broad distribution of randomly sampled languages, it provides a more comprehensive and reliable assessment of what these models can learn in practice. This addresses a significant limitation in prior work.

2. Another contribution is the development of novel algorithms for sampling random languages, particularly for Star-Free languages of bounded dot-depth and Context-Free languages. These methods are valuable and will facilitate more robust empirical studies in this research area.

3. The paper provides comprehensive preliminaries that effectively ground the study for a broad audience. The clear exposition of formal language theory, automata, and the specific language classes under investigation sets a solid foundation for understanding the experimental goals and results.

**Weaknesses:**

1. The discussion of the experimental results remains somewhat surface-level. For example, for Context-Free languages, where the paper notes the high average accuracy but low perfect learning rate, suggesting models only learn approximations. However, it does not delve deeper into why this might be the case or how the nature of the randomly sampled CFGs influences learnability. A more in-depth analysis connecting model failures to specific structural properties of the languages would significantly strengthen the paper's conclusions.

2. As an empirical study, the paper would benefit from providing more concrete details about the experimental setup. Key information, such as the size of the training/validation/test datasets for each language and the parameter counts of the different model architectures used, is missing from the main text. Including these details is crucial for assessing the scale of the experiments and the generalizability of the results.

3. The work is built upon, and frequently cites, a series of prior studies by Butoi et al. While grounding the work in established literature is good practice, the specific novel contributions of this paper could be more clearly introduced.

**Questions:**

My questions have been stated in detailed comments.

---

> ### Author Response · Authors · 2025-11-14
> **Response to reviewer rq1a**
>
> Thank you very much for your detailed review! We are glad that your found out methods clear and valuable. We address your comments in the following.
>
> > The discussion of the experimental results remains somewhat surface-level. For example, for Context-Free languages, where the paper notes the high average accuracy but low perfect learning rate, suggesting models only learn approximations. However, it does not delve deeper into why this might be the case or how the nature of the randomly sampled CFGs influences learnability. A more in-depth analysis connecting model failures to specific structural properties of the languages would significantly strengthen the paper's conclusions.
>
> Could you provide more details regarding the structural properties of the CFG that you mentioned?
>
> We are currently working on some more in-depth analysis for the context-free languages. We are planning to train an n-gram model to check whether the languages are inherently easier than, for instance, the regular languages, which would explain the better performance. Additionally, we are working on some analysis of the performance of the models as a function of string edit distance. Would this additional analysis strengthen our paper?
>
> > As an empirical study, the paper would benefit from providing more concrete details about the experimental setup. Key information, such as the size of the training/validation/test datasets for each language and the parameter counts of the different model architectures used, is missing from the main text. Including these details is crucial for assessing the scale of the experiments and the generalizability of the results.
>
> Thank you for your suggestion; we will include a summary in the main text and include more experimental details in the appendix.
>
> Currently, each dataset contains 10k training examples, 1k validation examples, and 5010 test examples, with approximately 10 examples for each length in the range [0,500]. We include experiments with various model sizes (128k, 256k, 512k). Table 1 reports the accuracy of the best-performing model across all of these configurations for expressivity and the mean across the best-performing configuration for inductive bias.
>
> We note that we are currently working on a set of new experiments with larger datasets, up to 100k examples, and slightly larger models.
>
> > The work is built upon, and frequently cites, a series of prior studies by Butoi et al. While grounding the work in established literature is good practice, the specific novel contributions of this paper could be more clearly introduced.
>
> Thank you for your suggestion. We will make sure to emphasize more our contributions in the introduction and well as in the respective sections. We will upload an updated version shortly.

---

> ### Author Response · Authors · 2025-11-26
> **Response to reviewer rq1a (2)**
>
> We ran some additional experiments and updated our draft to address your comments. We highlighted the changes in blue to make it easier for the reviewers to find these. You might be interested in the following changes:
>
> 1. We trained 3 non-neural baselines to assess the difficulty of the languages sampled from each language class, namely an n-gram model, an SVM and a logistic regression. We mention these in Sections 5 and 6 and give additional details in Appendix K. We observe the following: the performance of the baselines does not necessarily correlate with the complexity of the language class, or the performance of the neural models. Both the neural models and the non-neural baselines obtain very high accuracy on context-free languages, which might suggest that the sampled languages are inherently easier.
>
> 2. We added more concrete details about the experimental setup in Section 5 and added further details in Appendix F.
>
> 3. We rewrote parts of the Introduction and Related Work sections to hopefully make our contributions more clear. We also added some more comments that emphasize our contributions in the sections where we introduce them.
>
> We mention that we are still working on some new analysis for the context-free languages, e.g., analysis of the performance as a function of edit distance, and we will try to add these as soon as possible. Given that we made quite a few changes, we wanted to upload a new draft with the results we have so far so that the reviewers have enough time to check the changes before the end of the discussion period.

---

> > ### Author Response · Authors · 2025-11-30
> > **Response to reviewer rq1a (3)**
> >
> > > The discussion of the experimental results remains somewhat surface-level. For example, for Context-Free languages, where the paper notes the high average accuracy but low perfect learning rate, suggesting models only learn approximations. However, it does not delve deeper into why this might be the case or how the nature of the randomly sampled CFGs influences learnability. A more in-depth analysis connecting model failures to specific structural properties of the languages would significantly strengthen the paper's conclusions.
> >
> > We did some additional analysis to address reviewer rq1a's comment (see the discussion from Section 6 and Appendix L in the updated draft). In our original draft, we noticed that all architectures achieved around 95\% average accuracy on context-free languages. However, they mastered only a small subset of these languages perfectly, suggesting that while the models capture useful approximations, they still fail to learn the underlying generative mechanisms. To test this hypothesis, we plotted the cross-entropy of negative strings as a function of edit distance. We found that most misclassifications clustered at small edit distances, suggesting that the models rely on superficial cues rather than capturing the true structure of the target language. For comparison, we generated analogous plots for the star-free languages and observed that, in contrast, misclassifications in these languages are not correlated with edit distance.

---

### Official Review · Reviewer_uGxv · 2025-11-01

**Soundness:** 2
**Presentation:** 3
**Contribution:** 2
**Rating:** 6
**Confidence:** 2

**Summary:**

This paper empirically studies the expressiveness of several neural networks—Transformers, RNNs, and LSTMs. The paper tested their ability to learn randomly generated formal languages across the Chomsky hierarchy. The paper developed a probabilistic context-free grammar sampling algorithm to generate unbiased datasets, avoiding common hand-crafted benchmarks. The experiments reveal discrepancies between theoretical expressivity results and empirical behavior, particularly showing that Transformers underperform on some theoretically learnable languages, while RNNs and LSTMs sometimes exceed predicted expressivity.

**Strengths:**

1. The random language generation framework is novel and could become a useful tool for empirical expressivity research.
2. The evaluation spans multiple architectures and formal language classes with consistent methods.

**Weaknesses:**

1. The work identifies discrepancies but does not deeply investigate the reason behind it
2. The theoretical insight is borrowed from prior expressivity literature. There is no new formal result on understanding the expressiveness are introduced.
3. The sampled datasets and model sizes are relatively small, which limits generalizability.

**Questions:**

For transformer (TF) architecture, existing research works have pointed out that indeed TF or other neural nets cannot learn formal languages directly. However, under the concept or technique of chain-of-thought, TF models can solve difficult tasks to some extend. Can authors discuss how this work is related to the above observation?

---

> ### Author Response · Authors · 2025-11-14
> **Response to reviewer uGxv**
>
> Thank you very much for your review! We address your comments below.
>
> > The work identifies discrepancies but does not deeply investigate the reason behind it
>
> We assume that the discrepancies you mention are related to the context-free languages. Is this correct?
>
> We are currently working on some more in-depth analysis for the context-free languages. We are planning to train an n-gram model to check whether the languages are inherently easier than, for instance, the regular languages, which would explain the better performance. Additionally, we are working on some analysis of the performance of the models as a function of string edit distance. Would this additional analysis strengthen our paper?
>
> > The theoretical insight is borrowed from prior expressivity literature. There is no new formal result on understanding the expressiveness are introduced.
>
> Our paper does not seek to establish new theoretical results on the expressivity of neural networks. Instead, we aim to complement such theoretical work by empirically analyzing what different architectures can learn in practice, across languages belonging to various classes and subclasses of the Chomsky hierarchy. The motivation behind this approach is that theoretical analyses often rely on unrealistic assumptions that do not hold for real-world architectures.
> For example, while some work has shown that transformers are Turing-complete [1], these proofs depend on infinite-precision arithmetic, offering little insight into what transformers can actually learn in practice. Later work [2] demonstrated that transformers can express exactly the star-free languages, but these results also rest on assumptions such as hard attention, which are not implemented in practical models. Only very recent theoretical work [3] has adopted more realistic assumptions, showing that soft-attention transformers can learn only a subset of star-free languages, which are what we call PODFA languages in our paper. However, their experiments cover only a small number of languages, making it hard to assess the generalizability of their results. In contrast, our work evaluates transformers across many PODFA languages and empirically verifies their theoretical claims.
>
> Another key contribution of our paper is the introduction of algorithms for sampling random languages and random strings from several classes and subclasses of the Chomsky hierarchy. Such tools are often missing in empirical studies on the expressivity of neural architectures, which typically focus on a small set of handpicked languages. However, these limited examples are unlikely to be representative of an entire language class, making it difficult to draw robust conclusions about an architecture’s ability to learn that class. Our proposed sampling methods therefore provide an important resource for researchers seeking to empirically validate their theoretical claims.
>
> [1] Jorge Pérez, Pablo Barceló, and Javier Marinkovic.
> 2021. Attention is turing-complete. Journal of Machine Learning Research, 22(75):1–35.
>
> [2] Andy Yang, David Chiang, and Dana Angluin. Masked hard-attention transformers recognize exactly
> the star-free languages. In The Thirty-eighth Annual Conference on Neural Information Processing
> Systems, Vancouver, Canada, December 2024. URL https://openreview.net/forum?
> id=FBMsBdH0yz.
>
> [3] Jiaoda Li and Ryan Cotterell. Characterizing the expressivity of transformer language models, 2025.
> URL https://arxiv.org/abs/2505.23623.
>
> > The sampled datasets and model sizes are relatively small, which limits generalizability.
>
> We are currently running additional experiments with larger datasets, up to 100k training examples. We will come back to you shortly with the results.
>
> We tested models of various sizes, up to half a million parameters. Table 1 reports the best-performing models across all configurations. Importantly, increasing the model size did not consistently result in improved performance. However, we will run additional experiments with bigger models for some of the languages and report back.
>
> We will also include all these additional results in the appendix of the paper and upload a new draft.
>
>
> (continued in next comment)

---

> ### Author Response · Authors · 2025-11-14
> **Response to reviewer uGxv (2)**
>
> > For transformer (TF) architecture, existing research works have pointed out that indeed TF or other neural nets cannot learn formal languages directly. However, under the concept or technique of chain-of-thought, TF models can solve difficult tasks to some extend. Can authors discuss how this work is related to the above observation?
>
> Chain-of-thought reasoning has indeed been shown to increase the expressivity of transformers. However, this is beyond the scope of our paper, which focuses on the empirical expressivity of the standard transformer architecture without augmentations such as chain-of-thought. Even in the absence of such mechanisms, the practical capabilities of transformers remain poorly understood. While considerable theoretical work has attempted to formalize transformer expressivity, much of it relies on unrealistic assumptions that differ substantially from real-world models. At the same time, research on learnability demonstrates that certain functions are difficult for transformers to learn using standard optimization techniques such as stochastic gradient descent. For these reasons, we argue that it is essential to first analyze the practical abilities of standard transformers before considering improved versions such as those incorporating chain-of-thought reasoning.

---

> ### Author Response · Authors · 2025-11-26
> **Response to reviewer uGxv (3)**
>
> We ran some additional experiments and updated our draft to address your comments. We highlighted the changes in blue to make it easier for the reviewers to find these. You might be interested in the following changes:
>
> 1. We trained 3 non-neural baselines to assess the difficulty of the languages sampled from each language class, namely an n-gram model, an SVM and a logistic regression. We mention these in Section 5 and give additional details in Appendix K. We observe the following: the performance of the baselines does not necessarily correlate with the complexity of the language class, or the performance of the neural models. Both the neural models and the non-neural baselines obtain very high accuracy on context-free languages, which might suggest that the sampled languages are inherently easier.
>
> 2. We ran additional experiments with larger models and larger datasets for the PODFA and regular languages. We added a short summary of the results in Section 5 and added more details in Appendix I and Appendix J. We noticed that training on more data does improve performance in some cases. Thank you very much for suggesting this! The LSTM's accuracy on the regular languages increases from 82\% to 98\%. Both the transformer's and RNN's accuracies increase as well, but the gains are more modest. Increasing the model size, however, does not necessarily help. We noticed that the LSTM's performance improves slightly and the RNN's accuracy decreases as a function of model size.
>
> 3. We updated the introduction and related work sections to make our motivation and contributions more clear.
>
>
> We mention that we are still working on some new analysis for the context-free languages, e.g., analysis of the performance as a function of edit distance, and we will try to add these as soon as possible. Given that we made quite a few changes, we wanted to upload a new draft with the results we have so far so that the reviewers have enough time to check the changes before the end of the discussion period.

---

> > ### Author Response · Authors · 2025-11-30
> > **Response to reviewer uGxv (4)**
> >
> > > The work identifies discrepancies but does not deeply investigate the reason behind it
> >
> > We did some additional analysis to address reviewer uGxv's comment (see the discussion from Section 6 and Appendix L in the updated draft). In our original draft, we noticed that all architectures achieved around 95\% average accuracy on context-free languages. However, they mastered only a small subset of these languages perfectly, suggesting that while the models capture useful approximations, they still fail to learn the underlying generative mechanisms. To test this hypothesis, we plotted the cross-entropy of negative strings as a function of edit distance. We found that most misclassifications clustered at small edit distances, suggesting that the models rely on superficial cues rather than capturing the true structure of the target language. For comparison, we generated analogous plots for the star-free languages and observed that, in contrast, misclassifications in these languages are not correlated with edit distance.

---

### Author Response · Authors · 2025-12-03
**Final summary**

We want to thank the reviewers for their constructive feedback. Below we summarize the key updates and how they address the reviewers’ concerns. All changes are highlighted in blue in the revised draft.

1. **Additional analysis of context-free languages.** Reviewers uGxv and rq1a's requested more analysis of the context-free languages, given that our models' average classification accuracy was unexpectedly high (around 95\% average accuracy). One would expect that average accuracy would decrease as one goes higher in the Chomsky hierarchy. Indeed, when we look at the percentage of languages on which the models get 100\% accuracy (instead of accuracy averaged across languages), we do see this trend, as expected (Table 2, under Expressivity). In order to dig deeper into the high average accuracy on context-free languages, we have added an analysis of misclassified negative strings to Section 6 and Appendix L. We plot classification cross-entropy vs. the number of random edits applied when mutating positive strings into negative ones. We find that models struggle more on negative strings that have low numbers of edits, i.e., strings that look similar to positive ones. This pattern suggests that despite having high average accuracy, the models are still struggling near the decision boundary and have not mastered the underlying rules of the language, as expected. So, there does not seem to be a deficiency in the way the CFLs are generated, although it seems that the models are able to learn spuerficial heuristics that happen to get high (but imperfect) classification accuracy.
2. **Non-neural baselines.** In response to reviewers F4v3 and rgAY's comments about the difficulty of the sampled languages, we trained three non-neural baselines: an n-gram model, logistic regression, and an SVM (Section 5, Appendix K). We observed that the accuracy of these baseline models does not correlate cleanly with language-class complexity. On context-free languages, the baselines achieve 90\% accuracy, indicating again that superficial cues such as n-gram features are enough to get very good performance.
3. **Experiments with larger datasets and larger models.** Reviewer uGxv raised concerns about the relatively small datasets and model sizes. We trained models with up to 100k examples and increased model sizes up to approximately 1 million parameters (Section 5, Section 6, Appendices I, J). We mainly observed the following:
(1) Larger datasets improved performance in several cases: for example, the LSTM's accuracy on regular languages increased from 82\% to 98\%.
(2) Increasing model size did not uniformly help: the LSTM's accuracy improved slightly, while RNN's sometimes decreased.
4. **Clarification of our goal and contributions.** Reviewer uGxv noted that the paper does not introduce new theoretical expressivity results, and reviewer rq1a suggested that we should emphasize our contributions more. We clarified what our goal is in the introduction (Section 1): our contribution is intentionally empirical and complementary to theoretical work. We emphasize that many theoretical results rely on unrealistic assumptions that are not satisfied in practice, and our empirical evaluation bridges this gap. We also highlight our methodological contribution: algorithms for sampling random languages and strings from context-free languages, offering a useful resource that is largely missing from prior empirical studies. We have also added comments contrasting our paper to prior work in the Related Work section (Section 2) and in the sections where we introduce our methods. Finally, in response to reviewer's F4v3 comment, we have added a discussion about why our sampling methods lead to languages of interest.
5. **Title change.** Reviewer F4v3 pointed out that the title of our paper does not quite line up with the contribution of our paper. The purpose of our experiments is to show which languages neural networks can *learn* in practice, but our title characterized this as testing *expressivity* (whether a parameter setting for the language exists at all, regardless of reachability through training). We agree with this point and have changed our title accordingly, to reflect that we are interested in *learnability*, which provides a more holistic picture of neural networks' abilities than expressivity.
6. **Experimental details.** Reviewers F4v3 and rq1a expressed concerns that the original draft did not describe the experimental setup in enough detail. We have added more details in Section 5 of the main text and further details in appendix F.

---

### Meta-Review · Area_Chair_7D6z · 2026-01-06

**Summary:**

This work empirically tests the capabilities of deep neural networks in solving formal language tasks. It proposes algorithms to randomly sample formal languages from specific language classes and generate labeled datasets for training and evaluating neural networks on those languages.

**Reviewer Concerns:**

- The paper lacks sufficiently deep analysis of its empirical findings.
- The experimental evaluation is limited and lacks details, which undermines the generalizability and interpretability of the results and weakens support for the core empirical claims.
- This work does not introduce much new formal insights, and the paper’s novel contributions relative to existing work are unclear.

**Reviewer Scores:**

The original scores were 6/4/4/4. The major concern regarding the novelty and technical contributions of this work remains. Therefore, the reviewers would likely maintain their scores.

---

### Decision · Program_Chairs · 2026-01-26

Reject